# Development of a competition assay to assess the *in vitro* fitness of dengue virus serotypes using an optimized serotype-specific qRT-PCR

Anne-Fleur Griffon[1☉], Loeïza Rault[1☉], Clément Tanvet[1], Etienne Simon-Lorière[2], Myrielle Dupont-Rouzeyrol[1], Catherine Inizan[1]*

1 Dengue and Arboviruses, Research and Expertise Unit - Institut Pasteur in New Caledonia - Pasteur Network, Dumbéa-sur-Mer, New Caledonia, 2 Evolutionary genomics of RNA viruses Unit, Institut Pasteur, Université Paris Cité, CNRS UMR2000, Paris, France

☉ These authors contributed equally to this work.
* cinizan@pasteur.nc

## Abstract

### Background

Comparing the *in vitro* fitness of dengue virus (DENV) isolates is a pivotal approach to assess the contribution of DENV strains' replicative fitness to epidemiological contexts, including serotype replacements. Competition assays are the gold standard to compare the *in vitro* replicative fitness of viral strains. Implementing competition assays between DENV serotypes requires an experimental setup and an appropriate read-out to quantify the viral progeny of strains belonging to different serotypes.

### Methods

In the current study, we optimized an existing serotyping qRT-PCR by adapting primer/probe design and multiplexing the serotype-specific qRT-PCR reactions, allowing to accurately detect and quantify all four DENV serotypes. We next developed an *in vitro* competition assay to compare the replicative fitness of two DENV serotypes in the human hepatic cell line HuH7.

### Findings

The qRT-PCR was specific, and had a limit of detection below 7.52, 1.19, 3.48 and 1.36 genome copies/µL, an efficiency of 1.993, 1.975, 1.902, 1.898 and a linearity (R²) of 0.99975, 0.99975, 0.99850, 0.99965 for DENV-1, −2, −3 and −4, respectively. Challenge of this multiplex serotype-specific qRT-PCR on mixes of viral supernatants containing known concentrations of strains from two serotypes evidenced an accurate quantification of the amount of genome copies of each serotype. Quantification of the viral progeny of each serotype in the inoculum and the supernatant of competition assays using the serotype-specific multiplex qRT-PCR unveiled an enrichment

**Data availability statement:** All data generated or analyzed during this study are included in this paper and its Supporting Information files. Sequence data have been deposited in GenBank (https://www.ncbi.nlm.nih.gov/gen-bank) (MW315187, MW315194, PV791372-PV791376) and in the GISAID (https://gisaid.org/) EpiArbo database (EPI_SET_240904sd) (https://doi.org/10.55876/gis8.240904sd).

**Funding:** The current study received financial support from the Institut Pasteur "Actions Concertées Inter-Pasteuriennes" (ACIP-2019-281), from the Agence Nationale de la Recherche (ANR-19-CE35-0001-01-DENWOLUTION) and from the Fondation Ledoux-Jeunesse Internationale. The E.S.-L. laboratory is funded by Institut Pasteur, the INCEPTION program (Investissements d'Avenir grant ANR-16-CONV-0005), the Ixcore foundation for research, the French Government's Investissement d'Avenir programme, Laboratoire d'Excellence 'Integrative Biology of Emerging Infectious Diseases' (grant no. ANR-10-LABX-62-IBEID), the HERA Project DURABLE (grant no 101102733) and the NIH PICREID (grant no U01AI151758).

**Competing interests:** The authors declare that they have no competing interests.

**Abbreviations:** AU, Arbitrary Units; $CO_2$, Carbon dioxide; °C, Celsius degrees; CA, California; cDNA, Complementary DesoxyriboNucleic Acid; DENV, Dengue virus; DMEM, Dulbecco Modified Eagle Medium; FCS, Fetal Calf Serum; IFA, Immunofluorescent Focus Assay; FFU, Focus Forming Units; $H_2O$, Water; mL, milliliters; MOI, Multiplicity Of Infection; NCBI, National Center for Biotechnology Information; ONT, Oxford Nanopore Technologies; qPCR, quantitative Polymerase Chain Reaction; qRT-PCR, quantitative Real-Time Polymerase Chain Reaction; RNA, Ribonucleic Acid; TPB, Tryptose Phosphate Broth; UK, United Kingdom; USA, United States of America.

of the supernatant in DENV-1 genome copies, uncovering the enhanced replicative fitness of this DENV-1 isolate.

## Conclusions

This optimized qRT-PCR combined with a relevant cellular model allowed to accurately quantify the viral progeny of two DENV strains belonging to two different serotypes in a competition assay, allowing to determine which strain had a replicative advantage. This reliable experimental setup is adaptable to the comparative study of the replicative fitness of any DENV serotypes.

## Introduction

Dengue viruses are RNA viruses classified into four serotypes (DENV-1 to −4) based on their antigenic properties. DENV serotypes share between ~60–70% identity at the nucleotide sequence level [1,2]. The emergence of a new serotype has the potential to lead to outbreaks of enhanced breadth and severity, some DENV serotypes, genotypes or clades being associated with different severity, especially upon secondary infections [3]. However, the mechanisms underlying DENV serotypes co-circulation and replacements remain elusive. Viral replicative fitness may contribute to DENV serotype replacements. It is therefore of prime importance to set up experimental models to assess the relative replicative fitness of strains from different serotypes. For RNA viruses, an approximation to fitness is the relative ability to produce a stable infectious progeny in a given environment [4]. The study of viral replicative fitness has become an expanding field of research [5,6]. DENV fitness was initially approached through *in vitro* mono-infections of cellular models and quantification of the viral progeny or yield of virus, either by qRT-PCR or titration [7,8].

For many other viruses, the established gold standard to compare the replicative fitness is the setting up of a competition assay, which has allowed to compare the fitness of two variants of various viruses, including HIV-1 [5], influenza virus [5], Saint Louis Encephalitis virus [9], Bovine Viral Diarrhea Virus 1 and 2 [10] or mutants of Mayaro virus [11] for instance. The two viral variants or types were specifically quantified in the supernatant using either titration in the presence of variant-specific monoclonal antibodies [9], a flow cytometry assay using strain-specific fluorescent probes [10] or variant/type-specific primers and probe [9–11]. A search of PubMed using the following query ((dengue virus) AND ((competition) OR (coinfection)) AND ((cell) OR (cellular) OR (in vitro))) returned 259 results, among which many refer to competition and super-infection assays in cellular models between DENV and other arboviruses, including yellow fever virus [12], chikungunya virus [13], Sindbis Virus [14], Zika virus [15] or Japanese encephalitis virus and insect-specific viruses [16]. These assays were probed either with virus-specific qRT-PCR [13–15] or flow-cytometry using virus-specific antibodies [16]. Competition in mosquito C6/36 cells or adult mosquitoes between two DENV serotypes were also reported; they were probed using type-specific hyper-immune

mouse ascites [17,18], monoclonal antibodies [18] or a reverse transcription followed by a SybR Green qPCR using serotype-specific primers [19]. Finally, competition between two DENV NS4B mutants was reported and quantified by measurement of peak height at the polymorphic position in Sanger sequencing along with 3D-digital PCR with mutant-specific primers and probes [20].

Quantification by molecular techniques appears as the most time- and cost-efficient strategy to quantify the progeny yielded by strains belonging to two different serotypes. Several serotype-specific qRT-PCR have been developed to sero-type DENV [21,22]. Historically, dengue phylogenies mostly relied on E gene sequences [23]. As the E protein is a primary target for host immune pressure, it accumulates phylogenetically informative variations. Furthermore, E sequence varia-tion correlates with serotype/genotype structure, making it useful for inferring lineage/strain definitions. Therefore, several reliable DENV serotyping assays are based on E gene sequences [21,24]. The challenge is to leverage these techniques to allow the absolute quantification of the viral progeny of a cell culture simultaneously infected by strains belonging to two different serotypes. Furthermore, most competition assays implemented with DENV were developed in mosquito cells, recapitulating only part of the viral life cycle. In the current study, we aim to develop an *in vitro* competition assay in human cells to compare the relative replicative fitness of two DENV strains belonging to two different serotypes using a serotype-specific qRT-PCR read-out.

## Materials and methods

### Ethics statement

The current study was conducted in compliance with the Declaration of Helsinki principles. Serum samples collected in 1995−2019 came from laboratory-confirmed dengue patients who were informed of and did not object to the secondary use of their serum sample for research purposes. This study reusing serum samples received administrative and ethical clearance in France from the "Comité de Protection des Personnes Sud-Est II" (n° ID-RCB 2019-A03114-53, n° CPP 19.12.06.49357) and from the Consultative Ethics Committee of New Caledonia. The study was recorded on Clinicaltrials. gov (ID: NCT04615364).

### Human serum samples and viral isolates

Serum samples were collected for diagnosis purposes between 1995 and 2019 and stored in collection at −80°C at the territorial hospital. Twelve viral isolates obtained from these serum samples were used for the development of the qRT-PCR and the competition assay. Additionally, 8 isolates from arboviruses and an HCV clinical sample were used in a specificity assay and 87 serum samples were used for specificity/sensitivity determination, including five co-infections: DENV-1/DENV-2, DENV-1/DENV-4, DENV-2/DENV-3, DENV-2/DENV-4 and DENV-3/DENV-4. Serum leftovers were retrieved for the current study on July 17th, 2020 and April 3rd, 2025, (6–30 years after sample collection). RNA was extracted from viral isolates and serum samples using the QIAamp Viral RNA extraction kit (Qiagen, Hilden, Germany), following the manufacturer's instructions. Viral RNAs were stored at −80°C until subsequent use and subjected to detec-tion by qRT-PCR for the current study in 2021–2025.

### Cell culture

The *Aedes albopictus* C6/36 cell line was cultured at 28°C in Leibovitz L15 medium (Sigma-Aldrich, Merck, Steinheim, Germany) supplemented with 10% decomplemented Fetal Calf Serum (FCS, Gibco™, Fisher scientific, Paisley, UK) and 10% Tryptose Phosphate Broth (TPB, Gibco™, Fisher scientific, Paisley, UK). In humans, DENV mostly replicates in hepatocytes and cells from the monocytic lineage [25]. We therefore used the human hepatocarcinoma HuH7 cell line, which was cultured at 37°C under 5% $CO_2$ in Dulbecco's Modified Eagle Medium (DMEM, Gibco™, Fisher scientific, Pais-ley, UK) supplemented with 10% decomplemented FCS.

## Serotype and genotype determination

The serotype and genotype of DENV isolates used in the current study for qRT-PCR development and competition assays in HuH7 cells were determined by whole-genome sequencing performed on viral RNA extracted from viral cultures, using either an Illumina or an Oxford Nanopore Technologies (ONT) platform.

For sequencing on the Illumina platform, extracted RNA was treated with Turbo DNase (ThermoFisher, Asnières-sur-Seine, France) to digest contaminating cellular DNA. Host rRNA were depleted from RNA samples using the NEBNext® rRNA Depletion kit (New England Biolabs, Évry-Courcouronnes, France) as described previously [26]. RNA from selective depletion was used for cDNA synthesis and Illumina library preparation using the Nextera XT kit (Illumina) with dual indexes and sequenced on an Illumina NextSeq500 (75 cycles, paired-end reads) platform.

Raw paired-end files were processed for removal of Illumina adaptor sequences, trimmed and quality-based filtered using Trimmomatic v0.36 [27]. *De novo* assembly was performed using metaSPAdes v3.12.0 with default parameters [28]. Scaffolds were queried against the NCBI non-redundant protein database [29] using DIAMOND v 0.9.26 [30]. For each sample, the main scaffold corresponded to DENV and no other virus was identified. Iterative mapping using CLC-assembly-cell v5.1.0 was used to generate full-length or near full-length consensus genomes followed by manual curation when needed using Geneious Prime v2023. Whole-genome sequences were deposited on the GISAID (https://gisaid.org/) EpiArbo database (EPI_SET_240904sd https://doi.org/10.55876/gis8.240904sd).

For sequencing on the ONT platform, we used the tiling amplicons method for viral RNA which aims to prepare the extracted RNA into different ~1kb amplicons for whole genome sequencing. The first step involved cDNA synthesis through a Reverse Transcription using the Superscript III First Strand Synthesis kit (ThermoFisher Scientific, Carlsbad, CA, USA). Then, we performed a subsequent PCR step using the Q5 Hot Star High-Fidelity DNA polymerase (New England Biolabs, Ipswich, MA, USA) with 15–16 primers pairs, selected according to the serotype of the sample and designed in advance to generate amplicons that cover the entire genome. These PCR reactions were split in two pools of amplicons per sample. For each sample, after verifying the amplification with a Tapestation4150, the two pools of amplicons were purified using AMPure XP beads at a ratio of 1.8 and quantified with a Qubit2.0. We then combined the pools and prepared the sequencing library using the ONT Native Barcoding Kit24 V14 (SQK_NBD114.24). Sequencing was performed on a MinION Mk1D with a flow cell R10.4.1 (FLO-MIN114).

Raw POD5 files were processed for base-calling, demultiplexing, and ONT adapter sequence removal using Dorado/0.9.0 (https://github.com/nanoporetech/dorado). Sequencing quality control and analysis of the generated reads were conducted using FastQC/0.11.9 [31] and NanoPlot/1.44.0 (https://github.com/wdecoster/NanoPlot). *De novo* assembly was carried out using Canu/2.2 (https://github.com/marbl/canu), and the resulting contigs were blasted using BLAST+ (https://github.com/ncbi/blast_plus_docs). Reads were aligned to the reference genome with bwa/0.7.17 [32], and near full-length consensus genomes were generated using samtools/1.21 [33] and ivar/1.0.1 (https://github.com/andersen-lab/ivar). The consensus sequences were deposited in GenBank (accession numbers PV791372-PV791376). Table 1 legend indicates the accession numbers of whole-genome sequences generated in the current study. The serotype of clinical specimens used in the specificity/sensitivity assay was determined by the hospital diagnosis laboratory using its routine serotyping qRT-PCR [22].

## Phylogenetic analyses

A Maximum Likelihood phylogenetic tree showing the genotype/serotype of sequenced strains was built using MAFFT/7.525 [34] for sequence alignment, ensuring high accuracy in the alignment of nucleotides sequences. The resulting aligned sequences were then used as input for IQ-TREE/2.4.0 [35], which performed model selection and bootstrap analysis to infer the phylogenetic relationship among the strains, providing a robust framework for understanding their evolutionary history. The phylogenetic tree was visually edited using iTOL v7.2 [36]

**Table 1. Characteristics of the dengue viral isolates used in this study.**

| Serotype | Name of the strain | Year of collection | Concentration (genome copies/µL) | Titer (FFU/mL) | Used in monoplex qRT-PCR | Used for color compensation | Used in multiplex qRT-PCR | Used for standard curve | Used in mixes | Used in *in vitro* infections |
|---|---|---|---|---|---|---|---|---|---|---|
| DENV-1 | AVS73[a] | 2017 | | | X | X | | | | |
| | AVS94[b] | 2017 | $6.03 \times 10^5$ | $3.93 \times 10^7$ | | | X | X | X | X |
| DENV-2 | AVS109[c] | 2017 | | | X | X | | | | |
| | DWL089[d] | 2018 | $3.74 \times 10^5$ | $1.24 \times 10^7$ | | | X | X | X | X |
| DENV-3 | DWL133[d] | 1995 | | | X | | | | | |
| | 409[e] | 2008 | | | X | | | | | |
| | 240[d] | 2014 | | | X | X | | | | |
| | 5734[f] | 2014 | | | X | | | | | |
| | 5519[g] | 2014 | | | X | | | | | |
| | DWL039[d] | 2017 | $4.94 \times 10^5$ | $1.67 \times 10^7$ | X | | X | X | X | X |
| | DWL049[d] | 2017 | | | X | | | | | |
| DENV-4 | 8626[h] | 2009 | $1.36 \times 10^5$ | | X | X | X | X | | |

[a] GenBank accession number MW315187, [b] GenBank accession number MW315194, [c] GenBank accession number PV791376 [d] GISAID EPI_SET_240904sd https://doi.org/10.55876/gis8.240904sd, [e] GenBank accession number PV791372, [f] GenBank accession number PV791374, [g] GenBank accession number PV791373, [h] GenBank accession number PV791375.

## Production of DENV viral isolates

DENV isolates were recovered from DENV positive human serum samples through infection of C6/36 cells. All final viral stocks were prepared with no more than four passages in C6/36 cells for 5 days. Supernatants were collected and stored at −80 °C in 0.5 M Sucrose (Acros Organics, Geel, Belgium) and 20 mM Hepes buffer (Gibco, Fisher scientific, Paisley, UK). Viral isolates were titrated by Immunofluorescent Focus Assay (IFA) as described previously [37]; their titers were expressed in Focus Forming Unit per mL (FFU/mL). Viral isolates were recovered for 2 DENV-1 genotype I, 2 DENV-2 genotype Cosmopolitan, 7 DENV-3 genotype I and 1 DENV-4 genotype IIa, as determined by whole genome sequencing on the viral isolate retrieved from the serum sample.

## Quantification of the RNA concentration of DENV viral isolates by qRT-PCR

The absolute concentrations of DENV viral isolates were measured by Taqman qRT-PCR using the primers and the probe described in [38]. This qRT-PCR is pan-serotypic and allows the absolute quantification of DENV RNA copies number per µL. The reaction mix contained 0.3 µM of primers, 0.07 µM of FAM/BHQ-1-coupled probe, 1.2 µL of Superscript III Platinum One-Step enzyme (ThermoFisher Scientific, Carlsbad, CA, USA), 15 µL of the corresponding buffer (Invitrogen) and 5 µL of viral RNA extract in a reaction volume of 30 µL. Primers and probe sequences are given in Table 2. Thermal cycling protocol consisted in a reverse transcription step of 20 min at 50°C, followed by an enzyme activation and initial denaturation step of 2 min at 95°C, after which 50 cycles of 5 s of denaturation at 95°C and 30 s of annealing and extension at 60°C were implemented. Fluorescence was collected at the end of each annealing and extension step in the 465/510 nm channel. qRT-PCR reactions were performed using a LightCycler 480 (Roche, Basel, Switzerland). Absolute quantification of the number of copies genome per µL were performed using an RNA calibrator (TibMolBiol, Berlin, Germany). The calibrator sequence is given in the S1 File.

A standard curve was established by performing 10-fold dilutions of the RNA calibrator in water, from $10^8$ to $10^2$ copies/µL. Each dilution of the RNA calibrator was loaded in duplicate in the PCR mix described above and run using the thermal cycling protocol described above. The resulting standard curve was analyzed using the LightCycler 480

**Table 2. Sequences of the primers and probes and fluorochrome combinations for the pan-serotype absolute quantification of DENV [38] and for the optimized serotype-specific qRT-PCR.**

**Pan-serotype absolute quantification of DENV**

| Primer or probe | Sequence and fluorochromes from Warrilow et al. [38] | Regions of DENV genome amplified and detected |
|---|---|---|
| DenUniv UniF | AAG-GAC-TAG-AGG-TTA-KAG-GAG-ACC-C | 10578-10685 |
| DenUniv UniR | CGW-TCT-GTC-CCT-GGA-WTG-ATG | |
| FAM-DenUnivProbe TPLX-BHQ1 | FAM- TCT-GGT-CTT-TCC-CAG-CGT-CAA-TAT-GCT-GTT -TAMRA | |

**Optimized serotype-specific quantification of DENV**

| Serotype | Primer or probe | Initial sequences and fluorochromes from Ito et al. [21]. | Optimized sequences and fluorochromes | Regions of the E gene amplified and detected |
|---|---|---|---|---|
| DENV-1 | Forward primer D1MGBEn469s | GAA-CAT-GGR-ACA-AYT-GCA-ACY-AT | | 469-536 |
| | Reverse primer D1MGBEn536r | CCG-TAG-TCD-GTC-AGC-TGT-ATT-TCA | | |
| | Probe | FAM - ACA-CCT-CAA-GCT-CC – BHQ1 D1MGBEn493p | Atto425 – ACA-CCT-CAA-GCT-CC – BHQ1 Atto425-MxDEN-1P-BHQ1 | |
| DENV-2 | Forward primer D2MGBEn493s | ACA-CCA-CAG-AGT-TCC-ATC-ACA-GA | | 493-561 |
| | Reverse primer D2MGBEn561r | CAT-CTC-ATT-GAA-GTC-NAG-GCC | | |
| | Probe | Yakima Yellow - CGA-TGG-ART-GCT-CTC – BHQ1 D2MGBEn545p | Atto590 - CGA-TGG-ART-GCT-CTC - BHQ2 Atto590-MxDEN-2P-BHQ2 | |
| DENV-3 | Forward primer D3MGBEn1s | ATG-AGA-TGY-GTG-GGA-GTR-GGA-AAC | | 1-71 |
| | Reverse primer | CAC-CAC-DTC-AAC-CCA-CGT-AGC-T D3MGBEn71r | CAC-CAC-DTC-AAC-CCA-YGT-AGC-T DEN-3-eDyn | |
| | Probe D3MGBEn27p | ROX- AGA-TTT-TGT-GGA-AGG-YCT – BHQ2 | FAM - AGA-TTT-YGT-GGA-AGG-YCT – BHQ1 MxDEN-3Pdegenerated | |
| DENV-4 | Forward primer D4TEn711s | GGT-GAC-RTT-YAA-RGT-HCC-TCA-T | | 711-786 |
| | Reverse primer D4TEn786c | WGA-RTG-CAT-RGC-TCC-YTC-CTG | | |
| | Probe D4TEn734p | Cy5-CCA-AGA-GAC-AGG-ATG-TGA-CAG-TGC-TRG-GAT-C – BHQ2 | | |

software (S1 Fig). In each subsequent qRT-PCR, a sample containing $10^5$ genome copies/µL of the RNA calibrator (median concentration of RNA calibrator detected in the standard curve) was loaded and run in parallel of the RNA extracts to quantify. Concentrations were calculated using the standard curve previously established and the LightCycler 480 software.

## Optimization of a serotype-specific multiplex qRT-PCR

In order to quantify the absolute concentrations of DENV viral isolates belonging to two different serotypes in a mixture of two viral strains, we adapted and optimized a serotype-specific singleplex Taqman qRT-PCR [21]. This qRT-PCR was initially singleplex and serotype-specific and did not allow for the absolute quantification of DENV RNA copies number per μL for each serotype. Initial [21] and optimized primers and probes (Eurogentec, Seraing, Belgique) are detailed in Table 2.

The composition of multiplex reaction mixes in primers and probes, allowing the detection and quantification of all 4 DENV serotypes, is described in Table 3. Additionally, reaction mixes contained 0.8 μL of Superscript III Platinum One-Step enzyme (Invitrogen), 10 μL of the corresponding buffer (ThermoFisher Scientific, Carlsbad, CA, USA) and 4 μL of RNA extract in a reaction volume of 20 μL.

Thermal cycling protocol consisted in a reverse transcription step of 20 min at 50°C, followed by an enzyme activation and initial denaturation step of 2 min at 95°C, after which 50 cycles of 5 s of denaturation at 95°C and 30 s of annealing and extension at 60 °C were implemented. Fluorescence was collected at the end of each annealing and extension step in the 440/488, 498/580, 533/610 and 618/660 nm channels for DENV-1, DENV-3, DENV-2 and DENV-4, respectively. qRT-PCR reactions were performed using a LightCycler 480 (Roche, Basel, Switzerland). A color-compensation experiment was performed on the LightCycler 480 according to the manufacturer instructions. Briefly, for each of all 4 DENV serotypes, the RNA extract was loaded in pentaplicates in the corresponding qRT-PCR reaction mix containing all 8 primers and the probe specific to this serotype. The thermal cycling protocol consisted in a reverse transcription step of 20 min at 50°C, followed by an enzyme activation and initial denaturation step of 2 min at 95°C, after which 50 cycles of 5 s of denaturation at 95°C and 30 s of annealing and extension at 60 °C were implemented. The samples were then placed 1s at 95°C with a ramping of 4.4°C/s, then 30s at 40°C with a ramping of 2.2°C/s and underwent a ramping of 0.14°C/s up to 65°C during which a continuous acquisition of the fluorescence at each increase in 1°C was conducted in the 440/488, 498/580, 533/610 and 618/660 nm channels. Viral isolates of the four serotypes used for this color-compensation experiment are specified in Table 1. Following color compensation calculations, fluorescence detection thresholds of qRT-PCR results were manually adjusted based on the signal yielded by the serotype-specific positive controls (S2 Fig).

## Absolute quantification of the RNA concentrations of DENV viral isolates using the serotype-specific multiplex qRT-PCR

Viral isolates of the four serotypes used for this quantification are specified in Table 1. The absolute concentration of these RNA extracts was determined by qRT-PCR as described in a previous section. A standard curve was established by

Table 3. Final concentrations of primers and probes in the optimized serotype-specific qRT-PCR.

| Serotype | Primer or probe | Final concentration (μM) |
|---|---|---|
| DENV-1 | Forward primer D1MGBEn469s | 0.28 |
| | Reverse primer D1MGBEn536r | 0.28 |
| | Probe D1MGBEn493pa | 0.125 |
| DENV-2 | Forward primer D2MGBEn493s | 0.28 |
| | Reverse primer D2MGBEn561r | 0.28 |
| | Probe D2MGBEn545pa | 0.50 |
| DENV-3 | Forward primer D3MGBEn1s | 0.28 |
| | Reverse primer D3MGBEn71r | 0.28 |
| | Probe D3MGBEn27pa | 0.50 |
| DENV-4 | Forward primer D4TEn711s | 0.28 |
| | Reverse primer D4TEn786c | 0.28 |
| | Probe D4TEn734pb | 0.50 |

performing 10-fold dilutions of these RNA extracts in water. Each dilution of the RNA extract was loaded in duplicate in the PCR mix described and the thermal cycling protocol described above. The resulting standard curves were analyzed using the LightCycler 480 software (S3 Fig).

Absolute quantification of the number of genome copies per µL was then performed using these RNA extracts of known concentrations as calibrators: in each subsequent qRT-PCR, a sample of each of these in-house calibrators containing $5.08 \times 10^3$, $5.16 \times 10^3$, $7.14 \times 10^3$ and $1.36 \times 10^3$ genome copies/µL for DENV-1, DENV-2, DENV-3 and DENV-4, resp., was loaded and run in parallel of the RNA extracts to quantify. Concentrations were calculated in the LightCycler 480 software using the corresponding standard curve previously established.

## Specificity validation

Viral isolates of Zika virus (ZIKV) DAK 84 (African lineage), ZIKV MRS_OPY_Martinique (Asian lineage, American sub-lineage), ZIKV 5132 (Asian lineage, Pacific sub-lineage), chikungunya virus (CHIKV) NC/2011–568 (East-Central-South-African lineage), Ross River Virus from the French National Reference Center for arboviruses (RRV), West-Nile Virus from the French National Reference Center for arboviruses (WNV), the vaccine strain (Stamaril®) from Yellow Fever Virus (YFV), the vaccine strain (CD Jevax®) for Japanese Encephalitis (JEV) along with a serum sample positive for hepatitis C virus (HCV) were used in the specificity assay. Viral RNA was extracted from these samples using the QIAamp Viral RNA extraction kit (Qiagen, Hilden, Germany), following the manufacturer's instructions. Viral RNAs were stored at −80°C until subsequent use in the optimized serotype-specific qRT-PCR developed in the current study.

## Preparation of mixes of viral isolates from two different serotypes at a determined ratio

Viral isolates of known concentration were artificially mixed two by two to generate viral mixes containing 100:0, 75:25, 50:50, 25:75 and 0:100 ratios in genome copies of two DENV strains belonging to two distinct serotypes. Viral isolates used are specified in Table 1. Viral RNA was extracted from these artificial viral mixes as described in a previous section and concentrations in viral RNA from each serotype were quantified by the optimized serotype-specific multiplex qRT-PCR.

## Competition assay

HuH7 were seeded at $2 \times 10^5$ cells per well in 12-well plates. We assumed a doubling time of 24h for HuH7 cells [39], targeting $4 \times 10^5$ cells to be infected in each well 24h after seeding. Twenty-four hours later, sub-confluent cells were infected with DENV viral isolates diluted in DMEM supplemented with 2% FCS to a final volume of 300 µL. Cells were infected either in "mono-infections" by a single viral isolate or in "competition" by a mix of two viral isolates. Multiplicities of infection (MOI) were calculated for the $4 \times 10^5$ cells/well based on the titer of the viral isolate in FFU/mL determined by Immunofluorescent Focus Assay. For mono-infections, cells were infected with a single viral isolate at a MOI of 1. For competitions, cells were infected with a mix of two isolates belonging to two different serotypes at a MOI of 1 for each isolate (ratio 50:50) or at a MOI of 0.2 for one isolate and 1.8 for the other isolate (ratio 10:90). After two hours of incubation at 37°C under 5% $CO_2$, the inoculum was retrieved and stored at −80°C in 0.5 M Sucrose and 20 mM Hepes buffer for subsequent RNA extraction. Cells were washed twice with 1 mL of DMEM supplemented with 2% FCS. Cells were then cultured at 37°C under 5% $CO_2$ in 1 mL of DMEM supplemented with 2% FCS. At 72h post-infection, supernatants were collected and stored at −80°C in 0.5 M Sucrose and 20 mM Hepes buffer for subsequent RNA extraction. Experiments were performed in three independent replicates.

## Statistical analyses

Ninety-six RNA extracts, including 8 RNA extracts from arbovirus isolates, an HCV clinical sample and 87 serum samples from patients were analyzed by qRT-PCR in the sensitivity/specificity assay. A sample size above 95 gives a 99%

confidence to detect a 99% sensitivity/specificity ± 2% [40]. As a non-parametric test typically used for testing independence for 2 × 2 contingency tables, Fisher's exact test was used for testing the difference between observed and expected proportions in mixes of viral isolates or between the inoculum and the supernatant in competition assays. A *p* value < 0.05 was considered significant. Statistical analyses were performed using the GraphPad Prism Software 9.

## Results

### qRT-PCR optimization to detect all four DENV serotypes by qRT-PCR

The serotype and genotype of DENV strains used for qRT-PCR development were determined by whole-genome sequencing. Positioning of these strains in the context of the global DENV diversity are shown in a phylogenetic tree (S4 Fig). Serotypes are summarized in Table 1. We optimized an existing singleplex serotype-specific qRT-PCR [21] used for DENV serotyping in surveillance. Primers and probes from this qRT-PCR were coupled with a combination of fluorochromes compatible with multiplexing on the LightCycler 480 (Table 2): DENV-1 probe was coupled to FAM, DENV-2 probe to Yakima Yellow, DENV-3 probe to ROX and DENV-4 probe to Cy5 (Table 2). Alignment of these primers and probes with a set of published sequences (detailed in S1 Table) showed that primers and probes for the four DENV serotypes were located in the E gene (Fig 1A). In first intention, they were challenged in singleplex qRT-PCR for the accurate detection of DENV-1 to DENV-4 RNA extracted from viruses isolated in culture from serum samples collected in New Caledonia between 1995 and 2018 (Table 1). The DENV-1, −2 and −4 strains tested were accurately detected (S5 FigA, B and D). DENV-3 RNA extracted from viruses isolated in culture from serum samples collected in1995, 2008 and 2014 were also detected, albeit with low fluorescence signal (S5C Fig). DENV-3 strains retrieved in 2017 were not detected (S5C Fig).

We aligned DENV-3 primers and probes with DENV-3 sequences from genotype I, II and III including published sequences and sequences of DENV-3 genotype I strains retrieved in 2017 in New Caledonia generated in the framework of the current study. This alignment evidenced a perfect match for the forward primer (Fig 1B). However, this alignment revealed two mismatches with DENV-3 strains retrieved in New Caledonia in 2017 and in Malaysia: a mismatch at position 7 in the probe (C in DENV-3 VS T in the probe, Fig 1C) and a mismatch at position 16 in the reverse primer (A in the DENV-3 VS G in the reverse complemented sequence of the reverse primer, Fig 1D). Sequences of the probe and reverse primer were optimized according to these observations to include degenerated nucleotides able to detect the diversity of DENV-3 strains at these positions: a Y was introduced in the probe sequence at position 7, and a Y was introduced in the reverse primer sequence at position 16. Furthermore, the combination of fluorochromes was optimized for detection on the LightCycler 480 by coupling DENV-1 probe to Atto425, DENV-2 probe to Atto590, DENV-3 probe to FAM and DENV-4 probe to Cy5 (Table 2). Optimized primers and probes were challenged in singleplex qRT-PCR for the accurate detection of DENV-1 to DENV-4 RNA extracted from viruses isolated in culture from serum samples collected in New Caledonia between 1995 and 2018 (Table 1). All DENV-1 to −4 strains tested were accurately detected with a strong fluorescence signal (Fig 2A.-D.). Only DENV-2 fluorochrome Atto590 yielded a maximal fluorescence intensity of 0.64 AU (Fig 2B.).

### Multiplexing of the qRT-PCR to detect and quantify all four DENV serotypes

A color compensation experiment was implemented (S2 Fig). Optimized primers and probes were then challenged in multiplex qRT-PCR on DENV RNAs extracted from viruses isolated in culture from serum samples. Despite color compensation, minimal crosstalk was still observed between the DENV-1, −2, −3 and −4 channels, especially in the DENV-2 channel (Fig 3A.-D.). The fluorescence threshold was thus manually adjusted in the LightCycler 480 software, based on the signal of positive controls containing a single DENV strain of each DENV-1, −2, −3 and −4 serotype. This threshold adjustment allowed to detect the strains from each serotype in the dedicated channel. The threshold was adjusted in subsequent experiments using a similar approach based on positive controls. When the threshold was adjusted on the negative

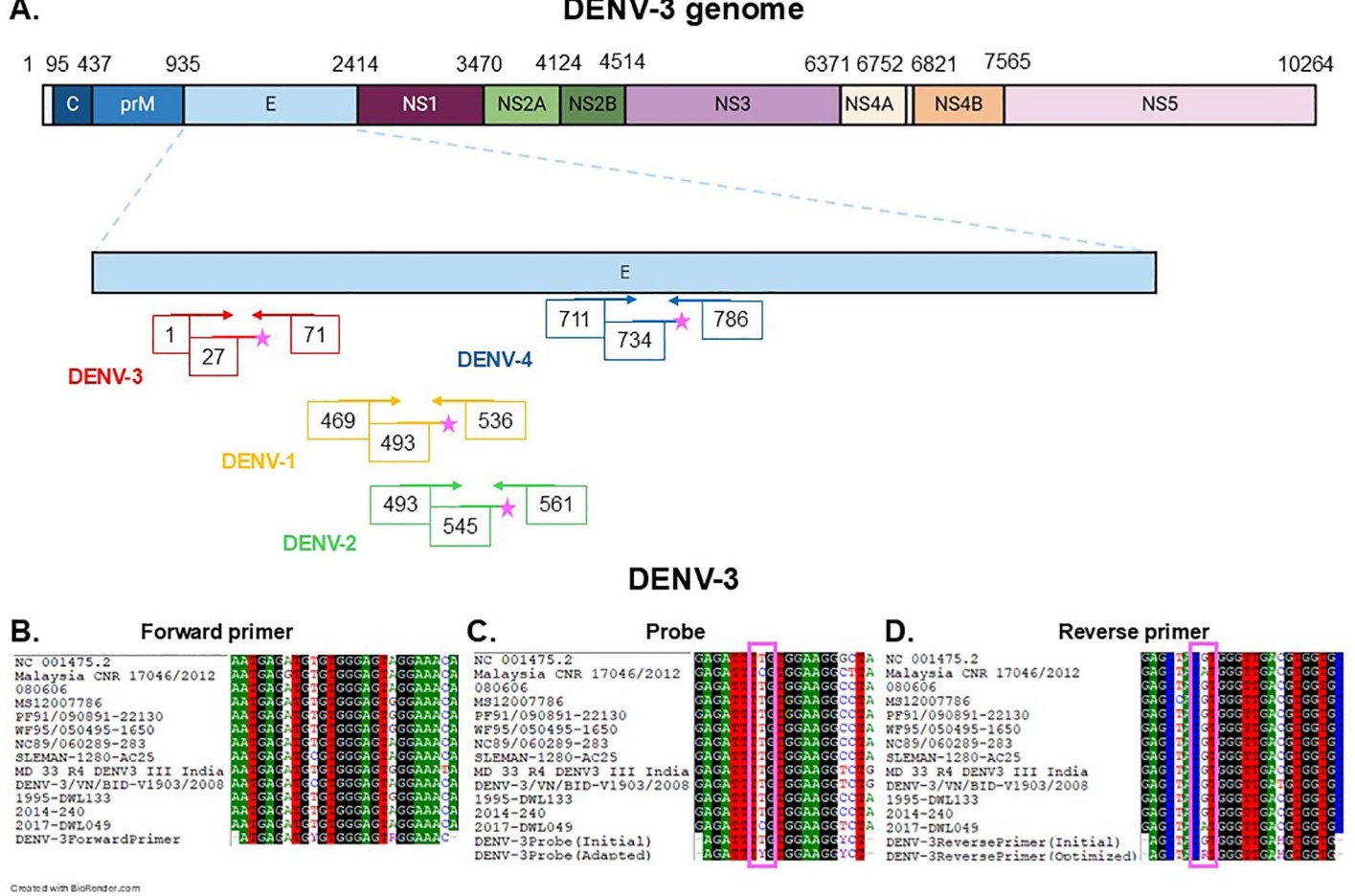

**Fig 1. Design of optimized primers and probes. A.** Positions of the 5' end of DENV-1 to −4 primers and probes on DENV-3 genome, as indicated in Table 2. Positions are given according to DENV-3 genome annotation (GenBank NC_001475.2). Pink stars represent the fluorochromes on the probes. **B-D.** Alignments of DENV-3 forward primer (**B.**), initial and adapted probe (**C.**) and initial and adapted reverse primer (**D.**) with DENV-3 sequences from genotypes I, II and III, including DENV-3 genotype I strains DWL133, 2014−240 and 2017-DWL049 retrieved in New Caledonia in 1995, 2014 and 2017 respectively. GenBank accession numbers of published DENV-3 sequences are given in S1 Table. Optimized positions are highlighted with a pink frame.

control, the average ΔCt value between the Ct of DENV-3 positive controls detected in the DENV-2 Atto590 channel and their Ct detected in the DENV-3 FAM channel across 25 independent experiments was 7.95±3.88.

Following color compensation and manual threshold adjustment, optimized primers and probes allowed the accurate detection of DENV-1 to DENV-4 in the multiplex qRT-PCR (Fig 3E.-H.). This qRT-PCR was therefore specific and accurately distinguished the four DENV serotypes. The coefficient of variation calculated for $5.08 \times 10^3$ copies/µL of DENV-1, $5.16 \times 10^3$ copies/µL of DENV-2 and $7.14 \times 10^3$ copies/µL of DENV-3 across 14 independent experiments was 6.34%, 2.17% and 2.29%, respectively.

The absolute concentration in genome copies per µL of four RNA extracts from the DENV-1 to −4 isolates were determined by absolute quantification using the qRT-PCR described in [38]. Using these four RNA extracts of known concentration, a standard curve was established for each DENV-1 to −4 serotypes (S3 Fig) as a serial ten-fold dilution of each RNAs performed in duplicate, allowing the absolute quantification of RNA extracts from DENV-1 to −4 with the multiplex serotype-specific qRT-PCR. This assay allowed to determine that the detection limit of this serotype-specific qRT-PCR

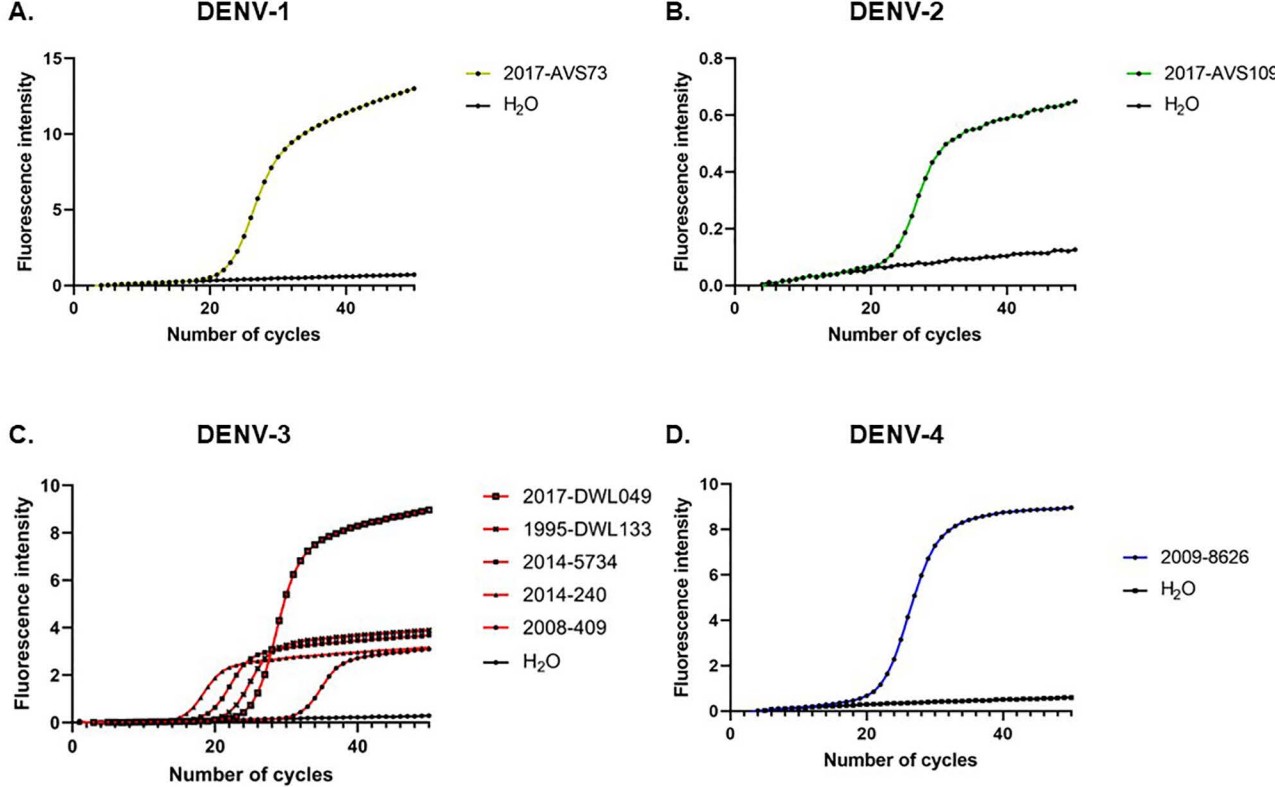

**Fig 2. Singleplex detection of DENV-1 to −4 with optimized primers and probes. A-D.** Fluorescence signal intensity as a function of the number of PCR cycles obtained with the optimized primers and probes are shown for DENV-1 (**A.**), DENV-2 (**B.**), DENV-3 (**C.**) and DENV-4 (**D.**).

was below 7.52, 1.19, 3.48 and 1.36 genome copies/µL for DENV-1, −2, −3 and −4 respectively (S3 Fig). The linear range of the assay extended from $7.52 \times 10^5$ to $7.52 \times 10^0$ copies/µL for DENV-1, $1.19 \times 10^6$ to $1.19 \times 10^1$ copies/µL for DENV−2, $3.48 \times 10^5$ to $3.48 \times 10^1$ copies/µL for DENV−3 and $1.36 \times 10^5$ to $1.36 \times 10^0$ copies/µL for DENV-4.

qRT-PCR amplification factor was 1.993, 1.975, 1.902, 1.898 for DENV-1, −2, −3 and −4, respectively. qRT-PCR efficiency was 99.3%, 97.5%, 90.2% and 89.8%. Linearity ($R^2$) was 0.99975, 0.99975, 0.99850, 0.99965 for DENV-1, −2, −3 and −4, respectively.

### Evaluation of the specificity and sensitivity of the multiplex serotype-specific qRT-PCR

To assess specificity, the fourplex serotype-specific qRT-PCR was tested against a panel of viruses including DENV-1 to −4, ZIKV, JEV, CHIKV, WNV, YFV, RRV and HCV. The DENV-1 to −4 RNAs were correctly detected by Atto425, Atto590, FAM and Cy5 signals (S6 Fig). The qRT-PCR did not generate any signal for ZIKV, JEV, CHIKV, WNV, YFV, RRV, and HCV, except for DENV-1, −2, −4 and YFV which generated residual signal with Ct ≥ 40 in the FAM channel, far above the detection limit of the fourplex qRT-PCR for DENV-3 which is ∼ 34 cycles as determined in the sensitivity assay (S6 Fig). The negative control without RNA template was also negative in the four channels.

Eighty-seven serum samples from DENV patients were included in a specificity/sensitivity assay, including 32 DENV-1, 20 DENV-2, 13 DENV-3, 7 DENV-4, one DENV-1/DENV-2, one DENV-1/DENV-4, one DENV-2/DENV-3, one DENV-2/DENV-4 and one DENV-3/DENV-4 co-infection and 10 negative samples according to the routine technique [22]. The optimized qRT-PCR correctly assigned 30 DENV-1, 19 DENV-2, 13 DENV-3 and 7 DENV-4 for a total of 69 correct

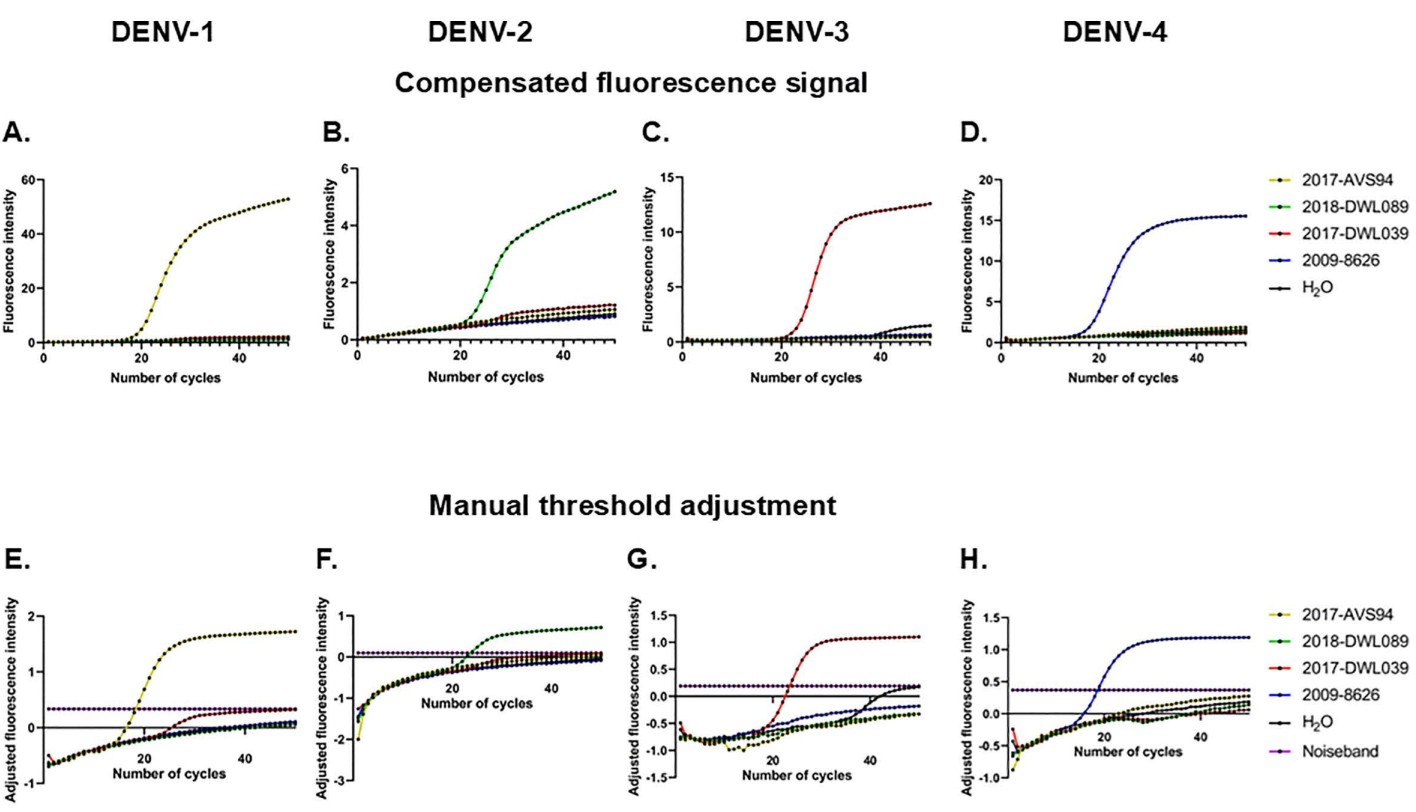

**Fig 3. Multiplex detection of DENV-1, −2, −3 and −4 using optimized primers and probes. A.-D.** Fluorescence signal intensity as a function of the number of PCR cycles after color compensation, for DENV-1 (**A.**), DENV-2 (**B.**), DENV-3 (**C.**) and DENV-4 (**D.**). **E.-H.** Fluorescence signal intensity as a function of the number of PCR cycles after color compensation and manual adjustment of the fluorescence signal, for DENV-1 (**E.**), DENV-2 (**F.**), DENV-3 (**G.**) and DENV-4 (**H.**).

assignments, no false positives, 9 samples identified as DENV-3 positive while being typed as negative by the routine technique and three false negatives, yielding 93.75% sensitivity, 100% specificity for DENV-1, 95% sensitivity, 100% specificity for DENV-2, 100% sensitivity, 86.95% specificity for DENV-3 and 100% sensitivity, 100% specificity for DENV-4. Co-infections were both reassessed by the routine serotyping technique historically used at the time of diagnosis [22] and assessed by the current optimized qRT-PCR. While the routine technique detected two co-infections (DENV-2/DENV-3 and DENV-2/DENV-4) out the five co-infections historically determined by the same technique at the time of sample collection, the current optimized qRT-PCR detected three co-infections (DENV-1/DENV-4, DENV-2/DENV-3, DENV-3/DENV-4).

Analysis of this specificity/sensitivity assay with threshold adjustment on the negative control and considering DENV-3 positive any sample with a Ct in the DENV-3 channel above 7.95 added to the Ct detected in DENV-2 channel correctly assigned 11/11 samples as compared to the routine technique. In addition, DENV-3 was identified as the infecting serotype in two samples identified as negative by the routine technique. Finally, analysis of the amplification curves (S7 Fig) allowed to identify DENV-3 as the infecting serotype in two samples identified as DENV-3 positive by the routine technique and 6 samples identified as negative by the routine technique. Only one sample typed as DENV-2 by the routine technique was identified as DENV-3 positive by our technique, yielding one false positive. Whole-genome sequencing of this strain could identify the correct assignment. Of note, the Ct for mono-infections did not notably increase with the duration since sample retrieval, indicating low viral RNA degradation in our collection of serum

samples. However, three co-infections could not be confirmed by the routine technique and two could not be confirmed by the current optimized qRT-PCR, nonetheless suggesting a partial RNA degradation impairing the appropriate detection of some co-infections.

## Quantification of DENV concentrations in a mix of two isolates belonging to different serotypes

The ability of the multiplex serotype-specific qRT-PCR to accurately quantify the concentration of two strains belonging to two different serotypes in the same RNA extract was next challenged. Concentration in viral genome copies per μL was thus determined for three viral isolates from DENV-1, −2 and −3 serotypes. These viral supernatants retrieved from infected C6/36 cultures were subsequently mixed two by two at ratios 0:100, 25:75, 50:50, 75:25 and 100:0. Following RNA extraction, the absolute concentration in viral genome copies per μL of each of these viral mixes was determined using the optimized quantitative multiplex serotype-specific RT-PCR. In each viral mix, the determined concentration of each strain as compared to its maximum in the 100:0 mix was not significantly different from the expected concentration (Fig 4A, $p > 0.05$ in Fisher's exact test for every determined concentration). Consistently, the determined relative percentage of each strain in the mix was not significantly different from the theoretical ratio introduced in the mix (Fig 4B, $p > 0.05$ in Fisher's exact test for every ratio).

## Comparison of the replicative fitness of two DENV serotypes

The relative replicative fitness of three DENV strains belonging to serotype −1, −2 and −3 was estimated through mono-infections of human HuH7 cells with a viral suspension containing a single viral isolate produced by infected C6/36 cells. As in humans, DENV mostly replicates in hepatocytes and cells from the monocytic lineage [25], the hepatocellular carcinoma cell line HuH7 was used as a physiological cellular model for DENV replication in its human host. The quantitative multiplex serotype-specific RT-PCR showed that the DENV-1 strain exhibited a higher concentration of genome copies per μL as compared to the DENV-2 and −3 strains, suggesting a higher replicative fitness (Fig 4C).

This replicative fitness was challenged in competition assays. HuH7 cells were infected with viral suspensions containing a ratio 50:50 of a combination of two DENV viral isolates from C6/36 cells, belonging to each of DENV-1, −2 and −3 serotypes. More specifically, the DENV-1 viral isolate AVS94 was mixed with either the DENV-2 viral isolate DWL-089 or the DENV-3 viral isolate DWL-039. The DENV-2 viral isolate DWL-089 was also mixed with the DENV-3 viral isolate DWL-039 (Table 1). Competition assays were performed in three independent replicates. Ratio calculation was based on the titer of viral suspensions determined by immunofluorescent focus assay. Concentrations of each viral isolate in the inoculum (Fig 4D) and the viral progeny at three days post-infection (culture supernatant, Fig 4E) were determined by applying the quantitative multiplex serotype-specific RT-PCR developed here. The supernatants of DENV-1:DENV-2 and DENV-1:DENV-3 mixes contained 97.5 and 60.5% of DENV-1 RNA copies/μL (Fig 4E) as compared to 14.8 and 9.9% in the inoculum (Fig 4D), respectively, evidencing a significant enrichment of the supernatant in DENV-1 RNA upon replication ($p < 0.001$ in Fisher's exact tests). Similarly, DENV-3 RNA was significantly enriched in the supernatant of cells infected with the DENV-3:DENV-2 mix (91.6% in the supernatant, Fig 4E VS 49.7% in the inoculum, Fig 4D, $p < 0.001$ in Fisher's exact test). This enrichment evidenced an enhanced replicative fitness of DENV-1 over DENV-2 and DENV-3, and of DENV-3 over DENV-2.

Competition assays performed with viral suspensions containing a ratio 10:90 of a combination of the same DENV strains isolated in C6/36 cells as the one used in the 50:50 competition assay evidenced a significant enrichment in DENV-1 in the supernatant (Fig 4G) of the DENV-1:DENV-2 mix as compared to the inoculum (Fig 4F, 41.2% VS 1.3%, $p < 0.001$ in Fisher's exact test), confirming DENV-1 enhanced replicative fitness over DENV-2. Similarly, DENV-3 was significantly enriched in the supernatant of DENV-3:DENV-2 mix (Fig 4G) as compared to the inoculum (Fig 4F, 29.8% VS 8.0%, $p < 0.001$ in Fisher's exact test), confirming DENV-3 fitness advantage over DENV-2. Only DENV-1 enhanced replicative fitness over DENV-3 was not confirmed in this 10:90 competition assay.

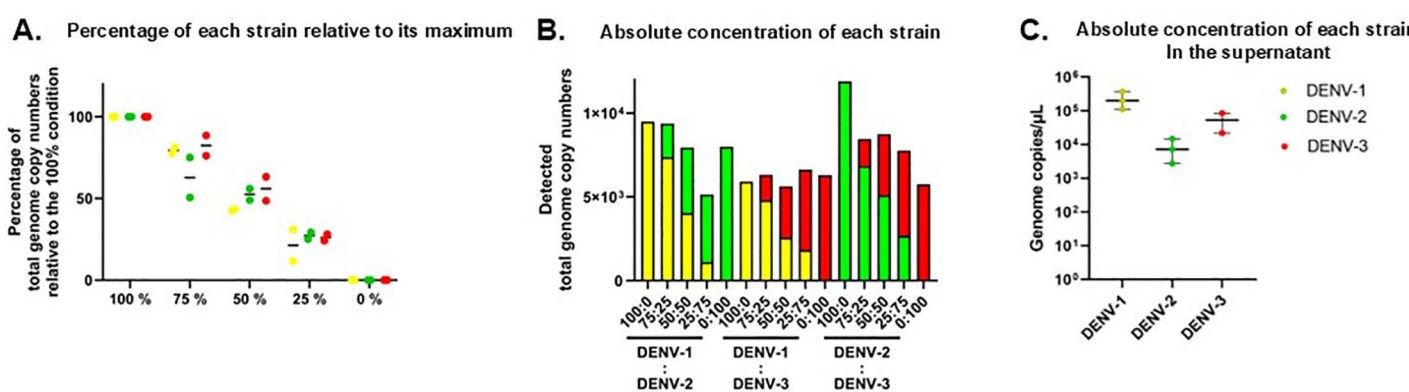

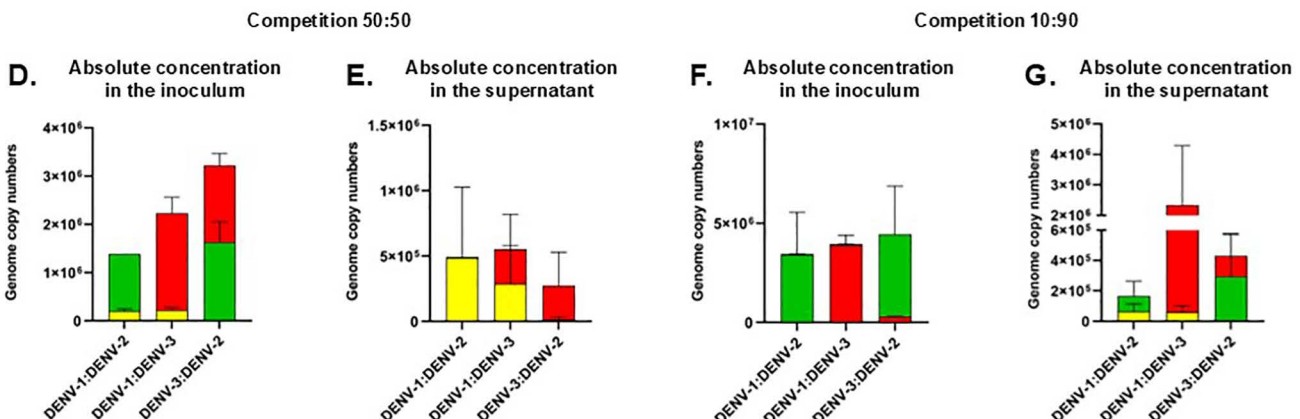

**Fig 4. Quantification of genome copies for each DENV serotype in mixes of viral suspensions.** Mixes of two viral suspensions at known ratios were subjected to RNA extraction and quantitative multiplex serotype-specific RT-PCR (**A-B.**). **A.** Viral RNA concentrations of the three strains from DENV-1 to −3 were expressed as a percentage of viral genome copy concentrations as compared to the maximum detected for each strain at a ratio 100:0. **B.** Relative RNA concentrations of the three strains from DENV-1 to −3 were expressed as a percentage of the total number of viral genome copy concentrations detected in each mix of two strains. **C.** Absolute quantification using the quantitative serotype-specific multiplex RT-PCR of DENV-1, −2 and −3 genome copies/μL in the supernatant of HuH7 infected with a viral suspension containing each a single viral isolate of each of the three serotypes (mono-infections). **D-E.** Absolute concentration of each isolate in the inoculum (**D.**) and the supernatant (**E.**) of HuH7 cells infected with a viral suspension containing a mix of DENV-1:DENV-2, DENV-1:DENV-3 or DENV-3:DENV-2 at a ratio 50:50 (competition assay). **F-G.** Absolute concentration of each isolate in the inoculum (**F.**) and the supernatant (**G.**) of HuH7 cells infected with a viral suspension containing a mix of DENV-1:DENV-2, DENV-1:DENV-3 or DENV-3:DENV-2 at a ratio 10:90 (competition assay). The concentration was determined using the quantitative serotype-specific multiplex RT-PCR. Mean and standard deviations of three independent triplicates are shown in **C-G**.

## Discussion

In the current study, we optimized a quantitative serotype-specific RT-PCR targeting DENV and developed a competition assay allowing to compare the fitness of strains belonging to two different DENV serotypes using this optimized qRT-PCR. This serotype-specific qRT-PCR is easily implementable in any laboratory equipped with a real time open-system thermocycler. This serotype-specific qRT-PCR offers rapid, sensitive and specific detection of DENV, allowing serotype differentiation and viral load quantification. Serotype differentiation is crucial for efficient epidemiological surveillance [41–43].

Quantifying specifically strains from different serotypes, genotypes or lineages can be based on Sanger sequencing, in which the height of the peak at the polymorphic position correlates with the amount of genome copies from the corresponding strain [20]. Next Generation Sequencing approaches may also be used, in which the number of reads of each strain correlates with the number of genome copies of the corresponding strain [44,45]. Both techniques are more expensive than a serotype-specific qRT-PCR and require both sequencing platforms and bioinformatics skills. For several years, numerous RT-PCR protocols have been developed to detect and differentiate dengue virus (DENV) serotypes in clinical samples, varying in the targeted genomic regions and detection strategies [46]. Multiplex approaches further enhanced efficiency by reducing reagent handling and support both serotype identification and viral load quantification in a single reaction tube [47]. Relying on the genetic divergence between DENV strains, including between strains of different genotypes or lineages, several multiplex qRT-PCR methods capable of serotyping have been developed as commercial kits such as RealStar® Dengue Type RT-PCR kit or custom in-house methods [48,49].

Serotyping dengue viruses could be used in vaccine development: the current optimized serotype-specific qRT-PCR could allow the relative quantification of vaccine strains from each of the four serotypes in a live attenuated vaccine, allowing to balance the inoculum and elicit an effective pan-serotype immunity. In addition, serotyping dengue viruses plays a key role in the epidemiological study of the disease [50,51]. Specific serotypes may be linked to different epidemics. Many studies have highlighted the importance of identifying events of DENV genotype and clade replacement, as seen, in the Americas [52], in Asia [53] and in the Pacific region [43]. In Brazil, the Ministry of Health therefore recommends that all suspected dengue samples be serotyped by PCR, enabling the evaluation of circulating serotypes [51]. A shift in the predominant serotype circulating within a community can signal an impending dengue outbreak, as the population may be immunologically naive to the new serotype. DENV serotype replacements are multifactorial, and herd immunity is an important contributor to the epidemiological dynamics of DENV serotypes, especially in non-endemic regions where an important fraction of the population may be susceptible to a serotype that has long been undetected locally. However, emergence or maintenance of DENV serotypes in some regions despite high herd immunity [37] and an increasing number of studies [6,54,55] suggest the contribution of virological factors in the epidemiological fitness of a given DENV strain belonging to an emerging genotype, lineage or clade, including its replicative fitness. Simultaneously, several studies report differences in viral load in patients according to the infecting DENV serotype [56–58], suggesting differences in replicative fitness between DENV serotypes. Their contribution to DENV serotype replacements remains elusive.

To improve dengue outbreak prevention but also the surveillance of antiviral resistance [59], it is nonetheless crucial to strengthen genomic surveillance [60] and better assess viral replicative fitness. For instance, a larger DENV-2 outbreak in Taiwan was associated with viral evolution since the previous outbreak. Acquisition of the NS5V357E mutation resulted in enhanced replicative fitness *in vitro* in mosquito and mammalian cells [6], highlighting the importance to assess viral replicative fitness. Similarly, emergence of the Asian I genotype of DENV-2 in Viet-Nam was associated with enhanced *in vivo* fitness denoted by a higher viraemia in patients [54]. Furthermore, enhanced replicative fitness in the mosquito vector was observed in New Caledonia upon a genotype replacement of DENV-1 strains [55]. A competitive index is a common method used in microbiology to assess the fitness and virulence of pathogenic organisms by comparing strains or mutants, helping to determine the role of specific genes under defined conditions [61]. *In vitro* competition fitness assays are essential tools for evaluating viral replication capacity, offering valuable insights into viral evolution, such as the role of HIV-specific mutations [62], while deepening our understanding of pathogenic mechanisms for adenoviruses for example [63]. These methods provide insights into replicative fitness and transmission between different hosts, for instance among emerging WNV strains [64], allow to unravel the mechanisms of competition and replacement of epidemic SARS-CoV-2 variants [65], and contribute to the optimization of treatment strategies [66], for instance for the monitoring of drug-resistant influenza strains [67,68]. These assays thus allow the monitoring of epidemiological dynamics to better inform public health interventions.

For DENV as well, competition assays allow to specifically detect significant differences in fitness between two different strains, belonging to two different serotypes for instance [17–19]. Importantly, a competition assay showed that the replicative fitness of an emergent DENV strain in Taiwan correlated with its epidemiologic and pathogenic potential [6], highlighting the epidemiological relevance of assessing DENV replicative fitness. In another instance, enhanced replicative fitness of primary viral isolates was attributed to a single amino-acid substitution in NS4B [20]. Both these competition assays compared the fitness of DENV variants differing by a few nucleotides. They were therefore assessed by Next-Generation Sequencing (NGS) approaches, which are laboratory intensive and costly. Here, we compared the fitness of strains belonging to different DENV serotypes, which differ by at least 30% nucleotide difference, using a competition assay revealed by an affordable and accessible qRT-PCR approach. The *in vitro* competition assay in human cells developed in the current study can now be implemented in studies of DENV serotype fitness in relevant epidemiological contexts. This competition assay design coupled to the serotype-specific qRT-PCR could be adapted to other human or mosquito cell lines. Additionally, fitness could be followed at different timepoints over time. Finally, this serotype-specific qRT-PCR could be implemented to quantify competition assays *in vivo* in the mosquito vector.

In our competition assay, although we infected the cells with equal or 10:90 MOIs of the two viral strains, the ratio in genome copies in the inoculum was different than 50:50 or 10:90. This discrepancy is due to differences in the ratio genome copies/FFU between strains, which produce variable amounts of defective viral particles, even from one viral preparation to the next [69]. This highlights the importance of adjusting the inoculum according to the MOI, to maintain a comparable load of infectious viruses in the inoculum. Further, performing infections with multiple ratios allows to confirm the observed trend.

This study suffers limitations. First, the accuracy of our serotype-specific quantification may be limited to the DENV lineages used to design the primers and probes. Only the sequences of DENV-3-specific primers and probes were adjusted. However, an alignment of their sequences with recent DENV-3 genomes collected in different locations (Fig 1) demonstrated that these sequences were likely adapted to detect all DENV-3 strains included in the sequence analysis. Second, the Atto590 fluorochrome used to detect and quantify DENV-2 showed a low signal. Signal-to-noise ratio could be enhanced using more efficient fluorochromes such as Alexa Fluor 594. Third, despite color compensation, we still observed some residual fluorescence cross-talk, which consequences were abrogated through a manual definition of the threshold based on the signal of positive controls. Fourth, the efficiency of our competition assay was not challenged for DENV-4. However, the study of DENV-4 was irrelevant in the recent epidemiology of dengue in New Caledonia and DENV-4 has a low circulation worldwide [70]. The strategy described in the current study can be easily adapted to assess DENV-4 replicative fitness, should this quantification be required in a given epidemiological context. Fifth, the current assay combined with the serotype-specific qRT-PCR we optimized does not allow to compare the replicative fitness of two strains belonging to two different genotypes within the same serotype. However, it can be easily adapted to compare the fitness within genotypes by applying genotype-specific qRT-PCR, as developed in [55]. Sixth, the competition assay developed in the current study requires a Biosafety Laboratory, which may be difficult to access in less favored areas. Finally, if the fitness of several strains from each serotype was to be quantified, the experimental setup can easily become laboratory intensive.

## Conclusions

Overall, we multiplexed and optimized a serotype-specific qRT-PCR to allow the simultaneous detection and quantification of the genome copies from two viral strains belonging to two different serotypes in a viral mixture. Coupled with a relevant competition assay described in the current study, this methodology allowed to compare the replicative fitness of strains belonging to two different serotypes. Such *in vitro* assessment of viral fitness may allow to evaluate the contribution of DENV replicative fitness to DENV serotype replacements in dengue-endemic or endemo-epidemic settings.

**Supporting information**

**S1 File. Sequence of the DENV RNA calibrator "iv-RNA 4".**
(DOCX)

**S1 Table. GenBank references of sequences used to design primers and probes of the optimized serotype-specific qRT-PCR.**
(DOCX)

**S1 Fig. Standard curve for the absolute quantification of DENV RNA genome copies using an RNA calibrator.** A. Compensated fluorescence intensity as a function of the number of PCR cycles for each dilution of the RNA calibrator. A signal was detected for the dilutions $10^8$ to $10^3$ copies/μL of the RNA calibrator. The limit of detection of this qRT-PCR was thus $10^3$ copies/μL. B. Number of PCR cycles as a function of the logarithmic transform of the concentration of the RNA calibrator. Technical duplicates were run simultaneously.
(TIF)

**S2 Fig. Color compensation.** A-D. Color compensation curves for DENV-1 (A.), DENV-2 (B.), DENV-3 (C.) and DENV-4 (D.). E.-H. Fluorescence signal intensity as a function of the number of PCR cycles after color compensation for DENV-1 (E.), DENV-2 (F.), DENV-3 (G.) and DENV-4 (H.).
(TIF)

**S3 Fig. Standard curves.** A-D. Fluorescence signal intensity as a function of the number of PCR cycles after color compensation and manual threshold adjustment on the negative control $H_2O$ for DENV-1 (A.), DENV-2 (B.), DENV-3 (C.) and DENV-4 (D.). Intersect of the X axis on the Y axis has been set at the threshold adjusted on the negative control. F-H. Standard curves for DENV-1 (E.), DENV-2 (F.), DENV-3 (G.) and DENV-4 (H.).
(TIF)

**S4 Fig. Phylogenetic trees of DENV strains used for qRT-PCR development.**
(PDF)

**S5 Fig. Singleplex qRT-PCR for the detection of DENV-1 to −4 with the initial primers and probes.** A-D. Fluorescence signal intensity as a function of the number of PCR cycles obtained with the initial primers and probes are shown for DENV-1 (A.), DENV-2 (B.), DENV-3 (C.) and DENV-4 (D.).
(TIF)

**S6 Fig. Specificity assay.** A-D. Fluorescence signal intensity as a function of the number of PCR cycles obtained for DENV-1 to −4, ZIKV, JEV, CHIKV, WNV, YFV, RRV and HCV in the DENV-1 (A.), DENV-2 (B.), DENV-3 (C.) and DENV-4 (D.) channels.
(PNG)

**S7 Fig. Example of curves in the DENV-2 and DENV-3 channels for equivocal samples.** Fluorescence signal intensity as a function of the number of PCR cycles after color compensation and manual threshold adjustment on the negative control $H_2O$ are shown for the DENV-2 (A.) and DENV-3 (B.) channels. Intersect of the X axis on the Y axis has been set at the threshold adjusted on the negative control. Round shapes indicate DENV-3 positive samples in the routine technique with a $\Delta Ct > 7.95$ between the DENV-2 and the DENV-3 channels in the optimized qRT-PCR, square shapes indicate DENV-3 positive samples in the routine technique with a $\Delta Ct < 7.95$ between the DENV-2 and the DENV-3 channels in the optimized qRT-PCR and triangle shapes indicate DENV-3 negative samples in the routine technique with a $\Delta Ct < 7.95$ between the DENV-2 and the DENV-3 channels in the optimized qRT-PCR.
(TIF)

 

## Acknowledgments

The authors thank Margot Petit, Dr Patrice Dunoyer and Dr Antoine Biron for technical assistance.

## Author contributions

**Conceptualization:** Etienne SIMON-LORIERE, Myrielle DUPONT-ROUZEYROL, Catherine Inizan.

**Formal analysis:** Anne-Fleur GRIFFON, Loeïza RAULT, Clément TANVET, Catherine Inizan.

**Funding acquisition:** Etienne SIMON-LORIERE, Myrielle DUPONT-ROUZEYROL, Catherine Inizan.

**Investigation:** Anne-Fleur GRIFFON, Loeïza RAULT, Clément TANVET, Catherine Inizan.

**Methodology:** Catherine Inizan.

**Project administration:** Catherine Inizan.

**Resources:** Myrielle DUPONT-ROUZEYROL.

**Supervision:** Myrielle DUPONT-ROUZEYROL.

**Validation:** Anne-Fleur GRIFFON.

**Visualization:** Anne-Fleur GRIFFON, Clément TANVET, Catherine Inizan.

**Writing – original draft:** Anne-Fleur GRIFFON, Catherine Inizan.

**Writing – review & editing:** Loeïza RAULT, Clément TANVET, Etienne SIMON-LORIERE, Myrielle DUPONT-ROUZEYROL.

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
