## [Decision Letter · Decision Letter 0]

7 Jan 2025

Dear Dr. Inizan,

Thank you for submitting your manuscript to PLOS ONE. After careful consideration, we feel that it has merit but does not fully meet PLOS ONE’s publication criteria as it currently stands. Therefore, we invite you to submit a revised version of the manuscript that addresses the points raised during the review process.

We look forward to receiving your revised manuscript.

Kind regards,

André Ricardo Ribas Freitas

Academic Editor

PLOS ONE

Journal Requirements:

2.  Please expand the acronym “NIH” (as indicated in your financial disclosure) so that it states the name of your funders in full.

“The current study received financial support from the Institut Pasteur “Actions Concertées Inter-Pasteuriennes” (ACIP-2019-281), from the Agence Nationale de la Recherche (ANR-19-CE35-0001-01-DENWOLUTION) and from the Fondation Ledoux-Jeunesse Internationale. The E.S.-L. laboratory is funded by Institut Pasteur, the INCEPTION program (Investissements d’Avenir grant ANR-16-CONV-0005), the Ixcore foundation for research, the French Government’s Investissement d’Avenir programme, Laboratoire d’Excellence ‘Integrative Biology of Emerging Infectious Diseases’ (grant no. ANR-10-LABX-62-IBEID), the HERA Project DURABLE (grant no 101102733) and the NIH PICREID (grant no U01AI151758).”

“The authors declare that they have no competing interests.”

5. In this instance it seems there may be acceptable restrictions in place that prevent the public sharing of your minimal data. However, in line with our goal of ensuring long-term data availability to all interested researchers, PLOS’ Data Policy states that authors cannot be the sole named individuals responsible for ensuring data access (http://journals.plos.org/plosone/s/data-availability#loc-acceptable-data-sharing-methods).

7. Please include captions for your Supporting Information files at the end of your manuscript, and update any in-text citations to match accordingly. Please see our Supporting Information guidelines for more information: http://journals.plos.org/plosone/s/supporting-information .

Additional Editor Comments:

Dear Dr. Catherine Inizan,

Thank you for submitting your manuscript titled "Development of a competition assay to assess the in vitro fitness of dengue virus serotypes using an optimized serotype-specific qRT-PCR" to PLOS ONE.

After careful evaluation by four reviewers, we regret to inform you that your manuscript requires major revisions before it can be considered for publication. Three reviewers have identified significant issues, especially with the methodology and the clarity of the results, which need to be addressed in detail. One reviewer recommended rejection. However, with substantial revisions, we believe the manuscript has the potential to be reconsidered for publication.

Please revise your manuscript to address all the comments and concerns raised by the reviewers. Specifically, focus on the following areas:

Methodology: Provide further clarification and detail regarding the experimental design and analysis. Reviewers highlighted the need for more robust justification for some of the methods used.

Data Interpretation: The presentation and interpretation of your results need to be clearer, particularly in how the data support your conclusions.

Additional Recommendations: Consider the minor recommendations provided by the reviewers to improve the clarity and readability of your manuscript.

Along with your revised manuscript, please include a detailed response letter addressing each of the reviewers' comments, explaining how you have modified the manuscript in response to their suggestions.

If you have any questions or need further assistance, please do not hesitate to contact me.

Best regards,

André Ricardo Ribas Freitas

Academic Editor

PLOS ONE

Reviewers' comments:

Reviewer's Responses to Questions

**Comments to the Author**

1. Is the manuscript technically sound, and do the data support the conclusions?

Reviewer #1: Yes

Reviewer #2: Partly

Reviewer #3: No

Reviewer #4: Partly

2. Has the statistical analysis been performed appropriately and rigorously?

Reviewer #1: N/A

Reviewer #2: Yes

Reviewer #3: I Don't Know

Reviewer #4: No

3. Have the authors made all data underlying the findings in their manuscript fully available?

Reviewer #1: Yes

Reviewer #2: No

Reviewer #3: Yes

Reviewer #4: Yes

4. Is the manuscript presented in an intelligible fashion and written in standard English?

Reviewer #1: Yes

Reviewer #2: Yes

Reviewer #3: Yes

Reviewer #4: Yes

Reviewer #1: The manuscript by Griffon and colleagues entitled “Development of a competition assay to assess the in vitro fitness of dengue virus serotypes using an optimized serotype-specific qRT-PCR” is interesting research.

In identifying DENV serotype is difficult, when there is a multiple serotype infection. Due to difficulty in identifying; most researchers report single serotype infection rather than multiple serotype infection. This research is quite useful. It will be much better, if the author has validated their claims with the serum sample containing multiple DENV serotypes. Below are the comments that will help to strengthen the manuscript.

Critical comments:

1. In order to avoid discrepancies related to suitability of using serum samples due to viral RNA degradation; it will be best to address the time span in which the samples were collected, stored and reused. Also indicate how these samples were stored in the methods section.

2. The authors have mentioned on the genotypes of DENV-1, 2, 3 & 4 for viral isolates. How the genotypes were identified and by using which gene? Are these from serum samples from your study? Whether these DENV strains were used in Huh7 cultures? Whether these strains were sequenced completely? Include these details in the manuscript.

3. There is no evidence of phylogenetic tree for the mentioned genotypes of DENV-1, 2, 3 & 4. Please provide details in methods and results section.

4. How many serum samples were used in this experiment? How long the serum samples were stored before RNA extraction? How these samples were stored? Indicate in which experiments these samples has been used? Include this information at appropriate places in methods section.

5. How the primers/probes were designed? Which gene was used for designing the primers/probes? Whether the designed primers/probes were from gene/genome sequences available from data repository or from your own sequencing or from other researcher’s publication? Please, include the details appropriately.

6. Competition assay – As per the methods, 2 x 105 HuH7 cells were seeded and after 24 hours viral isolates were introduced. Whether these cells attain confluence in 24 hrs? The cell population might also affect the titer value of viral load, right? How this was avoided?

7. The result section has no way to identify whether experimental observation is from human serum or cell culture. Please indicate wherever appropriate.

8. The competition assay shown interesting results. Whether the author validated in human serum samples? It is requested the author should validate the competition assay observation in their collected human serum samples containing one or more DENV serotypes.

9. Whether competition assay can help in identifying different genotypes within serotypes? Explain in the discussion section.

10. The discussion is superficial. It will be better to do an in-depth discussion. Compare your observation with other researchers’ article and discuss.

11. The authors have provided two genome data from GenBank. What is the purpose of providing it? If you used in your experiments, provide where and how. Similarly, you have provided data from GISAID, please include in which experiment you have used it or what is the importance of these sequence? Include at appropriate places.

12. Since the serum samples from before and after COVID pandemic i.e., from 1995-2021 was included in your study, did the author perform any COVID testing in these samples? If so, is there any difference in DENV viral load or serotype difference in COVID positive serum samples? It is advised to include COVID analysis status (as acknowledgement or wherever possible) in this manuscript which might be helpful for the researchers to know whether COVID has impact on DENV infection.

Specific comments:

Line 104-106: how the genotypes were identified? Provide details.

Line 114: Provide the primer/probe list as supplementary data.

Line 240-242: The sentence ended with a citation. From the information, it is a statement from your observation. Avoid providing citation for your statements unless you have already published article replated to this. It is suggested that the author should reframe the sentence.

Line 114 & 203: How many sets of primers/probes were used? The authors have mentioned two different articles (Warrilow et al., 2002, Ito et al., 2004) as reference. Clarify and rewrite.

Table 2: Lacks information on gene, gene location, repository name and ID from which the sequence was obtained.

The preprinted supporting methods section does not provide any valid information, it is just the repetition of the manuscript. Avoid submitting this or provide valid information for supporting additional methods which are not provided in the main manuscript.

Reviewer #2: This is an interesting manuscript but the authors need to make clear for the reviewers and all readers.

1. The authors described that the authors developed a new multiplex RT-PCR system for quantification of DENV for competition assay. Did the authors make validation about the cross reactivity f the primers and probes with other serotypes of the virus as well as the other flaviviruses. Please describe clearly at the revised manuscript.

2. Regarding the ethical issues, did the authors take written informed consent from the patients or please briefly describe about the justification for not taking ethical clearance.

3.The authors described the the concentration of the virus by mixing supernatants and the authors described that the results showed expected results. The authors should make confirmation of the results with another method such as immunofluorescence based focus assay method and the authors already established this system.

4. Regarding the competition assay, the authors must clear describe how many and how did the author check the competition assay such DENV1 and DENV-2 or DENV1 and DENV-3 or DENV2 and DENV-4. This point did not clear to the reviewers.

5. The authors did not describe how many samples were used in this study .

6. Why did the author used HuH cells for competitive assess and please briefly describe the justification.

7. For the competition assay, did the authors check different genotypes within one DENV serotypes. How did you do for two different genotypes competition assay because your RT-PCR system did not support for this competition assays,

Reviewer #3: This manuscript tried to introduce a novel competition assay of dengue virus serotypes using an optimized serotype-specific qRT-PCR. My major concerns are as follows:

1. The authors should provide more detailed results to support the specificity of the assays. From Figure 3, I could infer that the manual threshold adjustment is not reliable. Because fluorescence intensity could be influenced by many factors not restricted to the matching degree between the primers and the template. However, the delta Cp value between matched and unmatched primer pairs is relatively reliable. Recommend the authors set up a reliable threshold of the delta Cp value to support the specificity.

2. Could the author explain why the efficiencies of the qRT-PCR are so high and looks unnormal?

3. The authors should enroll more samples to validate the performance of the developed assays. and provide the related statistical analysis.

Reviewer #4: Introduction

- Line 44 - 45: DENV serotypes share between 34 and 73% identity at the nucleotide sequence level [Blok J et al., 1985].

Please check (https://doi.org/10.1186/s12864-019-6311-z). Inter-serotype nucleotide identity is around ~60-70% while polypeptide sequence identity might be as low as ~30%.

-Line 47-48: However, the mechanisms underlying DENV serotypes co-circulation and replacements are misunderstood.

To be misunderstood could mean "to fail to understand somebody/something correctly." Please consider using another word or rephrase the sentence.

-Line 48-51: Viral replicative fitness may contribute to DENV serotype replacements. It is therefore of prime importance to set up experimental models to assess the relative replicative fitness of strainsfrom different serotypes.

DENV serotype replacements could not be primarily attributed to replicative fitness. Host immune response and herd immunity might play more important roles in serotype replacements [https://doi.org/10.1073/pnas.1120621109]. Please provide more references in the introduction or the discussion to convince that replicative fitness evidently contribute to serotype replacements in human (or mosquito) population. If there is no evidence, please at least provide a use case for assaying replicative fitness in vitro that could contribute to the field (e.g. vaccine or drug development).

-Line 52-53: The study of viral replicative fitness has become an expanding field of research [Wargo et al., 2012].

The referenced paper was from 2012 (>10 years ago). Please provided a newer reference that the study of viral replicative fitness is still relevant today, especially in dengue research.

Materials and Methods

-Please add a "statistical analyses" part/sub-section that at least includes:

1) Sample size calculation (How do you justify that the number of samples or repeats are enough to accurately evaluate your assay?)

2) Describe statistical tests used in this study. If possible, explain why the tests were selected. For example, chi-square tests were used for testing the difference in proportions. Why Chi-square tests was preferred to Fisher's test?

3) Specify important statisitcal parameters. For example, the statistical significant level was set at 0.05 (i.e. p < 0.05).

4) Specify statistical software used in this study including version and company.

Results

-Khi² test (Line 260 and 263) and Chi² tests (Line 279, 285 and 288) should be changed to chi-squared tests or chi-square.

-Please report raw p-values in three decimal places (e.g. 0.001 - 0.009). The p-values could be reported as p<0.001 if they are smaller than 0.001.

-Since the authors propose a part this work as a quantification assay, the precision of the quantification must be presented. The coefficient of variation in a percentage form (%CV) is usually used for precision and repeatability evaluation of quantitative PCR assays. Please calculate %CV for all quantitation assays presented in this study.

-Line 245 - 247: This assay allowed to determine the sensitivity of this serotype-specific qRT-PCR, which was higher than 5.08x101, 5.16x101, 7.14x101 and 1.36x101 genome copies/µL for DENV-1, -2, -3 and -4 respectively.

What are these reported numbers? The lower limit of detection (LLOD) or the lower limit of quantification (LLOQ)? Please describe how these numbers were derived in the materials and methods. One way to calculate LLOD is to perform PCR with serially diluted samples with known concentrations in replicates (preferably >= 9 replicates). Then calculate a probability of detection curve by logistic regression. Finally, calculate a concentration (for dilution) that have a probability of detection = 95%. For LLOQ, %CV must be derived first. Then acceptable %CV must be specified (e.g. %CV <= 35%). Finally, the lowest concentration with %CV < acceptable %CV must be calculated for LLOQ. (LLOQ is the lowest concentration that could be repeatedly measured with a acceptable range of variations or errors).

- Comparison of the replicative fitness of two DENV serotypes and Figure 4

While the authors proposed this work for a quantification assay, the main results were reported as proportions (%), not absolute quantification results. The use of these assays might be limited since there already are several serotype-specific or pan-serotype quantification assays that report the quantification results in absolute forms (i.e. copies/ml or log10-copies/ml). In addtion, digital RT-PCR could directly report an absolute count of dengue genomes without standard curves.

The quantification of dengue virus with qRT-PCR could have a variation up to +/- 1 log10-copies so any difference < 1 log10-copies might just be an error of the quantification. It would be interesting to see the variation (or %CV) of the quantification assays proposed here.

Figure 4C is the only panel in Figure 4 that show results as absolute counts. However, the scale of the y-axis is misleading as the viral load has a skewed distribution. The y-axis should be changed to a log10-scale.

Also were experiments for comparing the replicative fitness performed in replicates? If so, please specify the number of replicates in the materials and methods. If there were no replicates, how are the error bars in Figure 4D and 4E derived?

Discussion

- Line 302-306: Of importance, this optimized multiplex serotype-specific qRT-PCR may be used in diagnostic laboratories to identify the infecting DENV serotype and could easily be deployed on the field using field-compatible real-time thermocyclers, albeit possibly requiring fluorochrome adjustments.

1) There is no need for serotyping in clinical settings. The doctors do not need to know serotype. Dengue-infected patients confirmed by PCR will be treated the same regardless of serotype.

2) There is no need for dengue quantification in epidemiological study. Serotyping results might be useful for surveillance as the replacement of the current predominant serotype may indicate the new outbreak cycle.

3) The quantification results are generally and only used in research.

4) The point-of-care PCR that could identify the serotype of dengue virus plus detecting other arboviruses (e.g. Zika virus, Chikungunya virus) is now commercially available for clinical use (https://www.sdbiosensor.com/product/product_view?product_no=23001).

Conclusion

Line 345-347: Such in vitro assessment of viral fitness may allow to evaluate the contribution of DENV replicative fitness to DENV serotype replacements in dengue-endemic or endemo-epidemic settings.

- Again, serotype replacements generally caused by host immune response and herd immunity. Please provide evidence that replicative fitness causes serotype replacements in real-life.

Figures

- There are no descriptions or annotations of all figures.

**Do you want your identity to be public for this peer review?** For information about this choice, including consent withdrawal, please see our Privacy Policy

Reviewer #1: No

Reviewer #2: No

Reviewer #3: **Yes: ** Jianguo Li

Reviewer #4: No

---

## [Author Response · Author response to Decision Letter 1]

23 Jul 2025

Please find below a point-by-point response to reviewers’ comments.

Additional Editor Comments:

Dear Dr. Catherine Inizan,

Thank you for submitting your manuscript titled "Development of a competition assay to assess the in vitro fitness of dengue virus serotypes using an optimized serotype-specific qRT-PCR" to PLOS ONE.

After careful evaluation by four reviewers, we regret to inform you that your manuscript requires major revisions before it can be considered for publication. Three reviewers have identified significant issues, especially with the methodology and the clarity of the results, which need to be addressed in detail. One reviewer recommended rejection. However, with substantial revisions, we believe the manuscript has the potential to be reconsidered for publication.

We thank the editorial team for this positive appraisal of our work.

Please revise your manuscript to address all the comments and concerns raised by the reviewers. Specifically, focus on the following areas:

Methodology: Provide further clarification and detail regarding the experimental design and analysis. Reviewers highlighted the need for more robust justification for some of the methods used.

Data Interpretation: The presentation and interpretation of your results need to be clearer, particularly in how the data support your conclusions.

Additional Recommendations: Consider the minor recommendations provided by the reviewers to improve the clarity and readability of your manuscript.

Along with your revised manuscript, please include a detailed response letter addressing each of the reviewers' comments, explaining how you have modified the manuscript in response to their suggestions.

If you have any questions or need further assistance, please do not hesitate to contact me.

Best regards,

André Ricardo Ribas Freitas

Academic Editor

PLOS ONE

Reviewer #1: The manuscript by Griffon and colleagues entitled “Development of a competition assay to assess the in vitro fitness of dengue virus serotypes using an optimized serotype-specific qRT-PCR” is interesting research.

In identifying DENV serotype is difficult, when there is a multiple serotype infection. Due to difficulty in identifying; most researchers report single serotype infection rather than multiple serotype infection. This research is quite useful. It will be much better, if the author has validated their claims with the serum sample containing multiple DENV serotypes. Below are the comments that will help to strengthen the manuscript.

Critical comments:

1. In order to avoid discrepancies related to suitability of using serum samples due to viral RNA degradation; it will be best to address the time span in which the samples were collected, stored and reused. Also indicate how these samples were stored in the methods section.

The qRT-PCR was optimized using viruses isolated from serum samples retrieved between 1995 and 2019. In compliance with the reviewer’s recommendation, we validated the qRT-PCR on clinical specimens collected between 1995 and 2019 and stored at -80°C before RNA extraction for the purpose of the current study. We did not observe RNA degradation. We modified the main text as follows:

Line 96-108:

Human serum samples and viral isolates

Serum samples were collected for diagnosis purposes between 1995 and 2019 and stored in collection at -80°C at the territorial hospital. Twelve viral isolates obtained from these serum samples were used for the development of the qRT-PCR and the competition assay. Additionally, 8 isolates from arboviruses and an HCV clinical sample were used in a specificity assay and 87 serum samples were used for specificity/sensitivity determination, including five co-infections: DENV-1/DENV-2, DENV-1/DENV-4, DENV-2/DENV-3, DENV-2/DENV-4 and DENV-3/DENV-4. Serum leftovers were retrieved for the current study on July 17, 2020 and April 3rd, 2025, (6-30 years after sample collection). RNA was extracted from viral isolates serum samples using the QIAamp Viral RNA extraction kit (Qiagen, Hilden, Germany), following the manufacturer’s instructions. Viral RNAs were stored at -80°C until subsequent use and subjected to detection by qRT-PCR for the current study in 2021-2025.

Line 408-410 “Of note, the Ct did not increase with the duration since sample retrieval, indicating the absence of viral RNA degradation in our collection of serum samples.”

2. The authors have mentioned on the genotypes of DENV-1, 2, 3 & 4 for viral isolates. How the genotypes were identified and by using which gene? Are these from serum samples from your study? Whether these DENV strains were used in Huh7 cultures? Whether these strains were sequenced completely? Include these details in the manuscript.

The serotype of DENV strains used in the current study has been determined either by whole-genome sequencing for strains used for qRT-PCR development and competition assays or by the serotyping qRT-PCR routinely used at the territorial hospital (Johnson et al., J Clin Microbiol, 2005) for clinical specimens. We have given details on the sequencing methods in the Methods section:

Line 121-167

Serotype and genotype determination

The serotype and genotype of DENV isolates used in the current study for qRT-PCR development and competition assays in HuH7 cells were been determined by whole-genome sequencing performed on viral RNA extracted from viral cultures, using either an Illumina or an Oxford Nanopore Technologies (ONT) platform.

For sequencing on the Illumina platform, extracted RNA was treated with Turbo DNase (ThermoFisher, Asnières-sur-Seine, France) to digest contaminating cellular DNA. Host rRNA were depleted from RNA samples using the NEBNext® rRNA Depletion kit (New England Biolabs, Évry-Courcouronnes, France) as described previously [23]. RNA from selective depletion was used for cDNA synthesis and Illumina library preparation using the Nextera XT kit (Illumina) with dual indexes and sequenced on an Illumina NextSeq500 (75 cycles, paired-end reads) platform.

Raw paired-end files were processed for removal of Illumina adaptor sequences, trimmed and quality-based filtered using Trimmomatic v0.36 [24]. De novo assembly was performed using metaSPAdes v3.12.0 with default parameters [25]. Scaffolds were queried against the NCBI non-redundant protein database [26] using DIAMOND v 0.9.26 [27]. For each sample, the main scaffold corresponded to DENV and no other viruses were identified. Iterative mapping using CLC-assembly-cell v5.1.0 was used to generate full-length or near full-length consensus genomes followed by manual curation when needed using Geneious Prime v2023. Whole-genome sequences were deposited on the GISAID (https://gisaid.org/) EpiArbo database (EPI_SET_240904sd doi: 10.55876/gis8.240904sd).

For sequencing on the ONT platform we used the tiling amplicons method for viral RNA which aims to prepare the extracted RNA into different ~1kb amplicons for whole genome sequencing. The first step involves cDNA synthesis through a RT using the Superscript III First Strand Synthesis kit (ThermoFisher Scientific, Carlsbad, CA, USA). Then, we performed a subsequent PCR step using the Q5 Hot Star High-Fidelity DNA polymerase (New England Biolabs, Ipswich, MA, USA) with 15 to 16 primers pairs, selected according to the serotype of the sample and designed in advance to generate amplicons that cover the entire genome. These PCR reactions were split in two pools of amplicons per sample. For each sample, after verifying the amplification with a Tapestation4150, the two pools of amplicons were purified using AMPure XP beads at a ratio of 1.8 and quantified with a Qubit2.0. We then combined the pools and prepared the sequencing library using the ONT Native Barcoding Kit24 V14 (SQK_NBD114.24). Sequencing was performed on a MinION Mk1D with a flow cell R10.4.1 (FLO-MIN114).

Raw POD5 files were processed for base-calling and demultiplexing, and ONT adapter sequence removal using Dorado/0.9.0 (https://github.com/nanoporetech/dorado). Sequencing quality control and analysis of the generated reads were conducted using FastQC/0.11.9 [28] and NanoPlot/1.44.0 (https://github.com/wdecoster/NanoPlot). De novo assembly was carried out using Canu/2.2 (https://github.com/marbl/canu), and the resulting contigs were blasted using BLAST+ (https://github.com/ncbi/blast_plus_docs). Reads were aligned to the reference genome with bwa/0.7.17 [29], and near full-length consensus genomes were generated using samtools/1.21 [30] and ivar/1.0.1 (https://github.com/andersen-lab/ivar). The consensus sequences were deposited in GenBank (accession numbers PV791372-PV791376). Table 1 legend indicates the accession numbers of whole-genome sequences generated in the current study. The serotype of clinical specimens used in the specificity/sensitivity assay was determined by the hospital diagnosis laboratory using its routine serotyping qRT-PCR [31].

3. There is no evidence of phylogenetic tree for the mentioned genotypes of DENV-1, 2, 3 & 4. Please provide details in methods and results section.

We built a Maximum Likelihood phylogenetic tree to position the DENV strains used in our study with regards to reported DENV genotypes/serotypes.

We have explained the phylogenetic analyses in the methods section and detailed the results of these phylogenetic analyses in the results section:

Line 168-174:

Phylogenetic analyses

A Maximum Likelihood phylogenetic tree showing the genotype/serotype of sequenced strains was built using MAFFT/7.525 [32] for sequence alignment, ensuring high accuracy in the alignment of nucleotides sequences. The resulted aligned sequences were then used as input for the IQ-TREE/2.4.0 [33], which performed model selection and bootstrap analysis to infer the phylogenetic relationship among the strains, providing a robust framework for understanding their evolutionary history. The phylogenetic tree was visually edited using iTOL v7.2 [34].

Line 306-309

The serotype and genotype of DENV strains used for qRT-PCR development have been determined by whole-genome sequencing. Positioning of these strains in the context of the global DENV diversity are shown in a phylogenetic tree (Supplementary figure 1). Serotypes are summarized in Table 1.

4. How many serum samples were used in this experiment? How long the serum samples were stored before RNA extraction? How these samples were stored? Indicate in which experiments these samples has been used? Include this information at appropriate places in methods section.

We used twelve viral isolates for technical development of the qRT-PCR and the competition assay and 8 viral isolates, an HCV clinical sample and 87 DENV clinical samples for specificity validation of the qRT-PCR. We have specified in the methods section how these samples have been used and how long serum samples were stored before RNA extraction. Table 1 summarizes the applications in which assay each of the twelve viral isolates have been used.

Line 96-108:

Human serum samples and viral isolates

Serum samples were collected for diagnosis purposes between 1995 and 2019 and stored in collection at -80°C at the territorial hospital. Twelve viral isolates obtained from these serum samples were used for the development of the qRT-PCR and the competition assay. Additionally, 8 isolates from arboviruses and an HCV clinical sample were used in a specificity assay and 87 serum samples were used for specificity/sensitivity determination, including five co-infections: DENV-1/DENV-2, DENV-1/DENV-4, DENV-2/DENV-3, DENV-2/DENV-4 and DENV-3/DENV-4. Serum leftovers were retrieved for the current study on July 17, 2020 and April 3rd, 2025, (6-30 years after sample collection). RNA was extracted from viral isolates serum samples using the QIAamp Viral RNA extraction kit (Qiagen, Hilden, Germany), following the manufacturer’s instructions. Viral RNAs were stored at -80°C until subsequent use and subjected to detection by qRT-PCR for the current study in 2021-2025.

5. How the primers/probes were designed? Which gene was used for designing the primers/probes? Whether the designed primers/probes were from gene/genome sequences available from data repository or from your own sequencing or from other researcher’s publication? Please, include the details appropriately.

Primers/probes developed for the current study were adapted from a published qRT-PCR from Ito et al. We specified it in the results section:

Line 309-313: We optimized an existing singleplex serotype-specific qRT-PCR [22] used for DENV serotyping in diagnostic laboratories. Primers and probes from this qRT-PCR were coupled with a combination of fluorochromes compatible with multiplexing on the LightCycler 480 (Table 2): DENV-1 probe was coupled to FAM, DENV-2 probe to Yakima Yellow, DENV-3 probe to ROX and DENV-4 probe to Cy5 (Table 2).

Primers and probes from all four DENV serotypes align to the E gene. We specified in the results section how we performed the alignments:

Line 313-316: “Alignment of these primers and probes with a set of published sequences (detailed in Supplementary table 1) showed that primers and probes for the four DENV serotypes are located in the E gene (Figure 1A).

And Line 324-335:

We aligned DENV-3 primers and probes with DENV-3 sequences from genotype I, II and III including published sequences and sequences of DENV-3 genotype I strains retrieved in 2017 in New Caledonia generated in the framework of the current study. This alignment evidenced a perfect match for the forward primer (Figure 1B). However, this alignment revealed two mismatches with DENV-3 strains retrieved in New Caledonia in 2017 and in Malaysia: a mismatch at position 8 in the probe (C in DENV-3 VS T in the probe, Figure 1C) and a mismatch at position 17 in the reverse primer (A in the DENV-3 VS G in the reverse complemented sequence of the reverse primer, Figure 1D). Sequences of the probe and reverse primer were optimized according to these observations to include degenerated nucleotides able to detect the diversity of DENV-3 strains at these positions: a Y was introduced in the probe sequence at position -8, and a Y was introduced in the reverse primer sequence at position 17.

6. Competition assay – As per the methods, 2 x 105 HuH7 cells were seeded and after 24 hours viral isolates were introduced. Whether these cells attain confluence in 24 hrs? The cell population might also affect the titer value of viral load, right? How this was avoided?

A seeding concentration 2x105 cells/well was determined in order to reach confluence at 24h. We considered a doubling time of 24h for HuH7 cells and assumed to reach 4x105 cells per well 24h after seeding. This number of cells was used to calculate the multiplicity of infection of 1. We specified in the Methods section:

Line 278-279: “HuH7 were seeded at 2 x 105 cells per well in 12-well plates. We assumed a doubling time of 24h for HuH7 cells [36], targeting 4 x 105 cells to be infected in each well 24h after seeding.”

Line 283-285: “Multiplicities of infection (MOI) were calculated for the 4 x 105 cells/well based on the titer of the viral isolate in FFU/mL determined by Immunofluorescent Focus Assay.”

7. The result section has no way to identify whether experimental observation is from human serum or cell culture. Please indicate wherever appropriate.

We specified the origin of the samples, either from serum sample or viral isolate, in any needed occurrence as follows:

Line 317-318 “RNA extracted from viruses isolated in culture from serum samples collected in”

Line 320-321 “DENV-3 RNA extracted from viruses isolated in culture from serum samples collected in”

Line 338-339 “DENV-1 to DENV-4 RNA extracted from viruses isolated in culture from serum samples collected”

Line 345-346 “on DENV RNAs extracted from viruses isolated in culture from serum samples.”

Line 363 “isolates”

Line 416 “isolates”

Line 416-417 “retrieved from infected C6/36 cultures”

Line 429 “single viral isolate produced by infected C6/36 cells.”

Line 436-437 “DENV viral isolates from C6/36 cells,”

Line 454 “DENV viral isolates from C6/36 cells”

8. The competition assay shown interesting results. Whether the author validated in human serum samples? It is requested the author should validate the competit

---

## [Decision Letter · Decision Letter 1]

29 Sep 2025

Dear Dr. Inizan,

Thank you for submitting your manuscript to PLOS ONE. After careful consideration, we feel that it has merit but does not fully meet PLOS ONE’s publication criteria as it currently stands. Therefore, we invite you to submit a revised version of the manuscript that addresses the points raised during the review process.

We look forward to receiving your revised manuscript.

Kind regards,

Julian Ruiz-Saenz

Academic Editor

PLOS ONE

Journal Requirements:

Reviewers' comments:

Reviewer's Responses to Questions

**Comments to the Author**

Reviewer #1: All comments have been addressed

Reviewer #2: All comments have been addressed

Reviewer #3: All comments have been addressed

Reviewer #4: (No Response)

2. Is the manuscript technically sound, and do the data support the conclusions?

Reviewer #1: Yes

Reviewer #2: Yes

Reviewer #3: Yes

Reviewer #4: Partly

3. Has the statistical analysis been performed appropriately and rigorously?

Reviewer #1: Yes

Reviewer #2: Yes

Reviewer #3: Yes

Reviewer #4: Yes

4. Have the authors made all data underlying the findings in their manuscript fully available?

Reviewer #1: Yes

Reviewer #2: Yes

Reviewer #3: Yes

Reviewer #4: Yes

5. Is the manuscript presented in an intelligible fashion and written in standard English?

Reviewer #1: Yes

Reviewer #2: Yes

Reviewer #3: Yes

Reviewer #4: Yes

Reviewer #1: The revised manuscript by Griffon and colleagues entitled “Development of a competition assay to assess the in vitro fitness of dengue virus serotypes using an optimized serotype-specific qRT-PCR” is much improved compared to the previous version. The authors’ efforts in revising the manuscript are commendable, and their responses to the queries are satisfactory.

Few things to consider…

Critical comments:

Why was the envelope (E) gene chosen for serotype-specific RT-PCR in DENV infection? Please provide a brief explanation in the Introduction section or a detailed explanation in the discussion section.

Specific comments:

Line 109-115: Please include the justification for using HuH7 cell line in this paragraph. This will help in understanding why HuH7 cells were used in the study.

Line 374-376: //… a DENV-1/DENV-2, a DENV-1/DENV-4, a DENV-2/DENV-3, a DENV-2/DENV-4 and a DENV-3/DENV-4//. Please replace ‘a’ with ‘1’.

Line 383-386: //Regarding the co-infections, while the routine technique detected two co-infections (DENV-2/DENV-3 and DENV-2/DENV-4) out the five co-infections historically determined at the time of sample collection, the current optimized qRT-PCR detected three co-infections (DENV-1/DENV-4, DENV-2/DENV-3, DENV-3/DENV-4)//. Since you describe the co-infections as ‘historically determined,’ does this mean they were previously reported by other researchers, or are they based on your current observations? Please clarify and include this information.

Line 394-395: // Only one sample typed as DENV-2 by the routine technique was identified as DENV-3 positive by our technique, yielding one false positive//. Do you think this discrepancy could be due to sequencing of a specific gene rather than the complete genome? Discuss…

Figure 4: Please provide color legend within the figure for D, E, F, & G.

Reviewer #2: Thank you very much for your great improvement of your manuscript and satisfactory revision. No more revision required.

Reviewer #3: I have no more questions, it seems that the authors do a good job and all my concerns have been addressed.

Reviewer #4: For a formatted version please see the attached file.

Overall:

The authors have addressed the technical concerns and revised the manuscript such that the methodological aspects now meet the standards of PLOS ONE. Nevertheless, I remain unconvinced of the broader applicability of the reported assay. Its utility appears to be restricted to laboratory-controlled co-infection experiments in cell culture, which represent highly artificial conditions. As such, the assay is unlikely to have meaningful relevance to serotype replacement or other processes of epidemiological importance in natural settings. To strengthen the manuscript, the authors may consider clarifying the potential practical applications of this assay, while avoiding references to serotype replacement caused by differences in viral replication levels. For example, could it be used to titrate each serotype in a live attenuated dengue vaccine so that the vaccine will induce balanced immune response against all four serotypes? (Again, antigenicity of each serotype is also important for balancing immune response.) The authors may focus more on co-infection study. For example, Quintero-Gil et al. (https://doi.org/10.3855/jidc.3978) evaluated replicative fitness of two DENV serotypes infecting a mosquito cell line and a live mosquito.

There are several studies reporting that each serotype of DENV intrinsically replicates at different rates. Tricou et al. (https://doi.org/10.1371/journal.pntd.0001309), Douyen at al. (https://doi.org/10.1093/infdis/jir014), and Voung et al. (https://doi.org/10.7554/eLife.92606.3) reported that DENV-1 shows higher early/febrile-phase viral loads than DENV-2 and DENV-3 in dengue patients. However, serotype replacement still occurs among all four serotypes. Therefore, the ability of viral replication may have limited relevance, or even be irrelevant, to the serotype replacement observed in epidemiological surveillance.

The authors try to convince that the difference in replicative fitness play a very important role in serotype replacement. However, the citations included actually demonstrated replication fitness changes that might contribute to “genotype”, “lineage” and “clade” replacement, not serotype replacement. Genotype, lineage and clade are more difficult to identify requiring virus genome sequencing or specialized PCR. The assay proposed by the authors cannot identify genotype of DENV or evaluate replicative fitness of genotypes within the same serotype in a co-infection. (The authors mention this in Line 544-547).

REF 6: Ko, H.-Y., et al., Emergence and increased epidemic potential of dengue variants with the NS5V357E mutation after consecutive years of transmission. iScience, 2024. 27(11): p. 110899. � The article reported an investigation of clade/strain replacement of DENV-2, not the serotype replacement.

REF 35: Inizan, C., et al., Viral evolution sustains a dengue outbreak of enhanced severity. Emerg Microbes Infect, 2021. 10(1): p. 536-544. � The article focuses on DENV-1 evolution (i.e., genotype or strain replacement), not serotype replacements.

REF 47: Gomgnimbou, M.K., et al., Utilization of novel molecular multiplex methods for the detection and, epidemiological surveillance of dengue virus serotypes and chikungunya virus in Burkina Faso, West Africa. Molecular Biology Reports, 2024. 51(1): p. 906. � The article does not address replicative fitness or provide any information regarding virus load quantification.

REF 48: O'Connor, O., et al., Potential role of vector-mediated natural selection in dengue virus genotype/lineage replacements in two epidemiologically contrasted settings. Emerg Microbes Infect, 2021. 10(1): p. 1346-1357. � The article investigated genotype/lineage replacements within a serotype, not serotype replacement.

REF 54: Vu, T.T., et al., Emergence of the Asian 1 genotype of dengue virus serotype 2 in Vietnam: in vivo fitness advantage and lineage replacement in South-East Asia. PLoS Negl Trop Dis, 2010. 4(7): p. e757 � The article investigated lineage replacement within DENV-2, not serotype replacement.

Abstract:

Line 27 – 29: The qRT-PCR was specific, and had a limit of detection below 7.52, 1.19, 3.48 and 1.36 genome copies/µL, an efficiency of 1.993, 1.975, 1.902, 1.898 and a linearity (R²) of 0.99975, 0.99975,0.9985, 0.99965 for DENV-1, -2, -3 and -4 respectively.

Please report R² values with the same/consistent decimal places.

Discussion:

Line 451- 453: This serotype-specific qRT-PCR offers rapid, sensitive, and specific detection of DENV, allowing serotype differentiation and viral load quantification, which are crucial for efficient epidemiological surveillance [39-41].

The authors state that both serotype differentiation and viral load quantification are essential for effective epidemiological surveillance. While serotype differentiation is indeed widely recognized and applied in epidemiological analyses, viral load quantification is not typically used for this purpose. Of the three references cited, all specifically demonstrate the application of serotype data in epidemiological studies, but none provide evidence for the use of viral load quantification in this context. Consequently, the statement in lines 451–453 is not entirely accurate. To resolve this, either “viral load quantification” in the sentence should be removed, or an additional citation must be provided that clearly demonstrates its role in epidemiological surveillance.

REF 39: Singh, K., et al., Identification of Dengue virus serotype and genotype: A comprehensive study from AIIMS Patna, Bihar. Indian Journal of Medical Microbiology, 2025. 53: p. 100789. � Serotyping was done, but viral load quantification was not done. This reference does not support that “viral load quantification” is crucial for epidemiological surveillance.

REF 40: Dieng, I., et al., Multifoci and multiserotypes circulation of dengue virus in Senegal between 2017 and 2018. BMC Infectious Diseases, 2021. 21(1): p. 867. � Serotyping was done, but viral load quantification was not done. This reference does not support that “viral load quantification” is crucial for epidemiological surveillance.

REF 41: Dupont-Rouzeyrol, M., et al., Epidemiological and molecular features of dengue virus type-1 in New Caledonia, South Pacific, 2001–2013. Virology Journal, 2014. 11(1): p.61. � Serotyping was done, but viral load quantification was not done. This reference does not support that “viral load quantification” is crucial for epidemiological surveillance.

Line 512 – 514: To improve dengue outbreak prevention and the surveillance of antiviral resistance [46], it is crucial to strengthen genomic surveillance but also to better assess viral replicative fitness [47].

REF 47: Gomgnimbou, M.K., et al., Utilization of novel molecular multiplex methods for the detection and, epidemiological surveillance of dengue virus serotypes and chikungunya virus in Burkina Faso, West Africa. Molecular Biology Reports, 2024. 51(1): p. 906. � The article does not address replicative fitness or provide any information regarding virus load quantification.

**Do you want your identity to be public for this peer review?** For information about this choice, including consent withdrawal, please see our Privacy Policy

Reviewer #1: No

Reviewer #2: No

Reviewer #3: **Yes: ** Jianguo Li

Reviewer #4: No

---

## [Author Response · Author response to Decision Letter 2]

12 Nov 2025

Please find below a point-by-point response to reviewers’ comments.

Reviewer #1: The revised manuscript by Griffon and colleagues entitled “Development of a competition assay to assess the in vitro fitness of dengue virus serotypes using an optimized serotype-specific qRT-PCR” is much improved compared to the previous version. The authors’ efforts in revising the manuscript are commendable, and their responses to the queries are satisfactory.

Few things to consider…

Critical comments:

Why was the envelope (E) gene chosen for serotype-specific RT-PCR in DENV infection? Please provide a brief explanation in the Introduction section or a detailed explanation in the discussion section.

Thank you for this suggestion. We have added to the introduction section, lines 81-86:

Historically, dengue phylogenies mostly relied on E gene sequences [23]. As the E protein is a primary target for host immune pressure, it accumulates phylogenetically informative variations. Furthermore, E sequence variation correlates with serotype/genotype structure, making it useful for inferring lineage/strain definitions. Therefore, several reliable DENV serotyping assays are based on E gene sequences [22, 24].

Specific comments:

Line 109-115: Please include the justification for using HuH7 cell line in this paragraph. This will help in understanding why HuH7 cells were used in the study.

We specified in the manuscript lines 119-123:

In humans, DENV mostly replicates in hepatocytes and cells from the monocytic lineage [25]. We therefore used the human hepatocarcinoma HuH7 cell line, which was cultured at 37°C under 5% CO2 in Dulbecco's Modified Eagle Medium (DMEM, Gibco™, Fisher scientific, Paisley, UK) supplemented with 10% decomplemented FCS.

Line 374-376: //… a DENV-1/DENV-2, a DENV-1/DENV-4, a DENV-2/DENV-3, a DENV-2/DENV-4 and a DENV-3/DENV-4//. Please replace ‘a’ with ‘1’.

We have modified lines 381-384:

Eighty-seven serum samples from DENV patients were included in a specificity/sensitivity assay, including 32 DENV-1, 20 DENV-2, 13 DENV-3, 7 DENV-4, one DENV-1/DENV-2, one DENV-1/DENV-4, one DENV-2/DENV-3, one DENV-2/DENV-4 and one DENV-3/DENV-4 co-infection and 10 negative samples according to the routine technique [34].

Line 383-386: //Regarding the co-infections, while the routine technique detected two co-infections (DENV-2/DENV-3 and DENV-2/DENV-4) out the five co-infections historically determined at the time of sample collection, the current optimized qRT-PCR detected three co-infections (DENV-1/DENV-4, DENV-2/DENV-3, DENV-3/DENV-4)//. Since you describe the co-infections as ‘historically determined,’ does this mean they were previously reported by other researchers, or are they based on your current observations? Please clarify and include this information.

These co-infections were “historically” determined by the diagnosis laboratory at the time of diagnosis. Probably due to sample degradation, the routine qRT-PCR historically used at the time of diagnosis and still in use by the diagnosis laboratory was not able to confirm these co-infections when recently run on the same samples preserved at -80°C.

We specified lines 390-395:

Co-infections were both reassessed by the routine serotyping technique historically used at the time of diagnosis and assessed by the current optimized qRT-PCR. While the routine technique detected two co-infections (DENV-2/DENV-3 and DENV-2/DENV-4) out the five co-infections historically determined by the same technique at the time of sample collection, the current optimized qRT-PCR detected three co-infections (DENV-1/DENV-4, DENV-2/DENV-3, DENV-3/DENV-4).

And added lines 405-410:

Of note, the Ct for mono-infections did not notably increase with the duration since sample retrieval, indicating low viral RNA degradation in our collection of serum samples. However, three co-infections could not be confirmed by the routine technique and two could not be confirmed by the current optimized qRT-PCR, nonetheless suggesting a partial RNA degradation impairing the appropriate detection of some co-infections.

Line 394-395: // Only one sample typed as DENV-2 by the routine technique was identified as DENV-3 positive by our technique, yielding one false positive//. Do you think this discrepancy could be due to sequencing of a specific gene rather than the complete genome? Discuss…

In the current manuscript, we describe the development of a qRT-PCR. Sequencing is indeed the method of choice to accurately confirm serotype assignment. Sequencing of the DENV strain contained in this serum sample could indeed confirm the actual infecting serotype. It is possible that due to sequence similarity, the current optimized serotype-specific PCR or the routine technique mistakenly assigned the wrong serotype to this putative atypical sequence. We added lines 404-405:

Whole-genome sequencing of this strain could identify the correct assignment.

Figure 4: Please provide color legend within the figure for D, E, F, & G.

We apologize for this mistake. We have added lines 655-662:

D-E. Absolute concentration of each isolate in the inoculum (D.) and the supernatant (E.) of HuH7 cells infected with a viral suspension containing a mix of DENV-1:DENV-2, DENV-1:DENV-3 or DENV-3:DENV-2 at a ratio 50:50. F-G. Absolute concentration of each isolate in the inoculum (F.) and the supernatant (G.) of HuH7 cells infected with a viral suspension containing a mix of DENV-1:DENV-2, DENV-1:DENV-3 or DENV-3:DENV-2 at a ratio 10:90. The concentration was determined using the quantitative serotype-specific multiplex RT-PCR. Mean and Standard Deviations of three independent triplicates are shown in D-G.

Reviewer #2: Thank you very much for your great improvement of your manuscript and satisfactory revision. No more revision required.

We thank the Reviewer for this positive appraisal.

Reviewer #3: I have no more questions, it seems that the authors do a good job and all my concerns have been addressed.

We thank the Reviewer for this positive feed-back.

Reviewer #4: For a formatted version please see the attached file.

Overall:

The authors have addressed the technical concerns and revised the manuscript such that the methodological aspects now meet the standards of PLOS ONE. Nevertheless, I remain unconvinced of the broader applicability of the reported assay. Its utility appears to be restricted to laboratory-controlled co-infection experiments in cell culture, which represent highly artificial conditions. As such, the assay is unlikely to have meaningful relevance to serotype replacement or other processes of epidemiological importance in natural settings. To strengthen the manuscript, the authors may consider clarifying the potential practical applications of this assay, while avoiding references to serotype replacement caused by differences in viral replication levels. For example, could it be used to titrate each serotype in a live attenuated dengue vaccine so that the vaccine will induce balanced immune response against all four serotypes? (Again, antigenicity of each serotype is also important for balancing immune response.)

We thank the reviewer for this interesting and innovative suggestion. We have added to the discussion section lines 484-487:

Serotyping dengue viruses could be used in vaccine development: the current optimized serotype-specific qRT-PCR could allow the relative quantification of vaccine strains from each of the four serotypes in a live attenuated vaccine, allowing to balance the inoculum and elicit an effective pan-serotype immunity.

The authors may focus more on co-infection study. For example, Quintero-Gil et al. (https://doi.org/10.3855/jidc.3978) evaluated replicative fitness of two DENV serotypes infecting a mosquito cell line and a live mosquito.

We thank the Reviewer for this identification of another study analyzing DENV serotypes replicative fitness in competition assay. We have added this reference in the existing relevant paragraph lines 73-76:

Competition in mosquito C6/36 cells or adult mosquitoes between two DENV serotypes were also reported; they were probed using type-specific hyper-immune mouse ascites [17, 18], monoclonal antibodies [18] or a reverse transcription followed by a SybR Green qPCR using serotype-specific primers [19, 20].

And to the discussion section lines 526-527:

For DENV as well, competition assays allow to specifically detect significant differences in fitness between two different strains, belonging to two different serotypes for instance [17-20].

There are several studies reporting that each serotype of DENV intrinsically replicates at different rates. Tricou et al. (https://doi.org/10.1371/journal.pntd.0001309), Douyen at al. (https://doi.org/10.1093/infdis/jir014), and Voung et al. (https://doi.org/10.7554/eLife.92606.3) reported that DENV-1 shows higher early/febrile-phase viral loads than DENV-2 and DENV-3 in dengue patients. However, serotype replacement still occurs among all four serotypes. Therefore, the ability of viral replication may have limited relevance, or even be irrelevant, to the serotype replacement observed in epidemiological surveillance.

We thank the reviewer for this insightful comment. We have added lines 502-505:

Simultaneously, several studies report differences in viral load in patients according to the infecting DENV serotype [56-58], suggesting differences in replicative fitness between DENV serotypes. Their contribution to DENV serotype replacements remains elusive.

The authors try to convince that the difference in replicative fitness play a very important role in serotype replacement. However, the citations included actually demonstrated replication fitness changes that might contribute to “genotype”, “lineage” and “clade” replacement, not serotype replacement. Genotype, lineage and clade are more difficult to identify requiring virus genome sequencing or specialized PCR. The assay proposed by the authors cannot identify genotype of DENV or evaluate replicative fitness of genotypes within the same serotype in a co-infection. (The authors mention this in Line 544-547).

REF 6: Ko, H.-Y., et al., Emergence and increased epidemic potential of dengue variants with the NS5V357E mutation after consecutive years of transmission. iScience, 2024. 27(11): p. 110899. � The article reported an investigation of clade/strain replacement of DENV-2, not the serotype replacement.

REF 35: Inizan, C., et al., Viral evolution sustains a dengue outbreak of enhanced severity. Emerg Microbes Infect, 2021. 10(1): p. 536-544. � The article focuses on DENV-1 evolution (i.e., genotype or strain replacement), not serotype replacements.

REF 47: Gomgnimbou, M.K., et al., Utilization of novel molecular multiplex methods for the detection and, epidemiological surveillance of dengue virus serotypes and chikungunya virus in Burkina Faso, West Africa. Molecular Biology Reports, 2024. 51(1): p. 906. � The article does not address replicative fitness or provide any information regarding virus load quantification.

REF 48: O'Connor, O., et al., Potential role of vector-mediated natural selection in dengue virus genotype/lineage replacements in two epidemiologically contrasted settings. Emerg Microbes Infect, 2021. 10(1): p. 1346-1357. � The article investigated genotype/lineage replacements within a serotype, not serotype replacement.

REF 54: Vu, T.T., et al., Emergence of the Asian 1 genotype of dengue virus serotype 2 in Vietnam: in vivo fitness advantage and lineage replacement in South-East Asia. PLoS Negl Trop Dis, 2010. 4(7): p. e757 � The article investigated lineage replacement within DENV-2, not serotype replacement.

We agree with the reviewer that we should clarify the fact that the studies we refer to considered genotype/lineage/clade replacements and not serotype replacements. We have specified lines 498-505:

However, emergence or maintenance of DENV serotypes in some regions despite high herd immunity [39] and an increasing number of studies [6, 55, 56] suggest the contribution of virological factors in the epidemiological fitness of a given DENV strain belonging to an emerging genotype, lineage or clade, including its replicative fitness. Simultaneously, several studies report differences in viral load in patients according to the infecting DENV serotype [57-59], suggesting differences in replicative fitness between DENV serotypes. Their contribution to DENV serotype replacements remains elusive.

We have also deleted lines 482-483 in the discussion section:

In recent years, understanding the role of genotypes and clades in DENV serotype replacements, as well as genotype- and clade-specific immunity, has become increasingly crucial.

Abstract:

Line 27 – 29: The qRT-PCR was specific, and had a limit of detection below 7.52, 1.19, 3.48 and 1.36 genome copies/µL, an efficiency of 1.993, 1.975, 1.902, 1.898 and a linearity (R²) of 0.99975, 0.99975,0.9985, 0.99965 for DENV-1, -2, -3 and -4 respectively.

Please report R² values with the same/consistent decimal places.

We apologize for this mistake and have corrected the R² for DENV-3 to 0.99850, both in the abstract and line 371.

Discussion:

Line 451- 453: This serotype-specific qRT-PCR offers rapid, sensitive, and specific detection of DENV, allowing serotype differentiation and viral load quantification, which are crucial for efficient epidemiological surveillance [39-41].

The authors state that both serotype differentiation and viral load quantification are essential for effective epidemiological surveillance. While serotype differentiation is indeed widely recognized and applied in epidemiological analyses, viral load quantification is not typically used for this purpose. Of the three references cited, all specifically demonstrate the application of serotype data in epidemiological studies, but none provide evidence for the use of viral load quantification in this context. Consequently, the statement in lines 451–453 is not entirely accurate. To resolve this, either “viral load quantification” in the sentence should be removed, or an additional citation must be provided that clearly demonstrates its role in epidemiological surveillance.

REF 39: Singh, K., et al., Identification of Dengue virus serotype and genotype: A comprehensive study from AIIMS Patna, Bihar. Indian Journal of Medical Microbiology, 2025. 53: p. 100789. � Serotyping was done, but viral load quantification was not done. This reference does not support that “viral load quantification” is crucial for epidemiological surveillance.

REF 40: Dieng, I., et al., Multifoci and multiserotypes circulation of dengue virus in Senegal between 2017 and 2018. BMC Infectious Diseases, 2021. 21(1): p. 867. � Serotyping was done, but viral load quantification was not done. This reference does not support that “viral load quantification” is crucial for epidemiological surveillance.

REF 41: Dupont-Rouzeyrol, M., et al., Epidemiological and molecular features of dengue virus type-1 in New Caledonia, South Pacific, 2001–2013. Virology Journal, 2014. 11(1): p.61. � Serotyping was done, but viral load quantification was not done. This reference does not support that “viral load quantification” is crucial for epidemiological surveillance.

We thank the reviewer for this suggestion. We have modified lines 464-466 as follows:

This serotype-specific qRT-PCR offers rapid, sensitive, and specific detection of DENV, allowing serotype differentiation and viral load quantification. Serotype differentiation is crucial for efficient epidemiological surveillance [42-44].

Line 512 – 514: To improve dengue outbreak prevention and the surveillance of antiviral resistance [46], it is crucial to strengthen genomic surveillance but also to better assess viral replicative fitness [47].

REF 47: Gomgnimbou, M.K., et al., Utilization of novel molecular multiplex methods for the detection and, epidemiological surveillance of dengue virus serotypes and chikungunya virus in Burkina Faso, West Africa. Molecular Biology Reports, 2024. 51(1): p. 906. � The article does not address replicative fitness or provide any information regarding virus load quantification.

We thank the reviewer for his/her carefully reading of this section. We moved the reference upstream in the sentence for ac

---

## [Decision Letter · Decision Letter 2]

1 Dec 2025

Development of a competition assay to assess the in vitro fitness of dengue virus serotypes using an optimized serotype-specific qRT-PCR

PONE-D-24-55413R2

Dear Dr. Inizan,

We’re pleased to inform you that your manuscript has been judged scientifically suitable for publication and will be formally accepted for publication once it meets all outstanding technical requirements.

Kind regards,

Julian Ruiz-Saenz, PhD

Academic Editor

PLOS ONE

Additional Editor Comments (optional):

Reviewers' comments:

Reviewer's Responses to Questions

**Comments to the Author**

Reviewer #1: All comments have been addressed

Reviewer #4: All comments have been addressed

2. Is the manuscript technically sound, and do the data support the conclusions?

Reviewer #1: Yes

Reviewer #4: Yes

3. Has the statistical analysis been performed appropriately and rigorously?

Reviewer #1: Yes

Reviewer #4: Yes

4. Have the authors made all data underlying the findings in their manuscript fully available?

Reviewer #1: Yes

Reviewer #4: Yes

5. Is the manuscript presented in an intelligible fashion and written in standard English?

Reviewer #1: Yes

Reviewer #4: Yes

Reviewer #1: The authors have addressed all the queries raised.

Only a few English grammar and syntax errors remain in the manuscript; these should be rectified before publication.

Reviewer #4: (No Response)

**Do you want your identity to be public for this peer review?** For information about this choice, including consent withdrawal, please see our Privacy Policy

Reviewer #1: No

Reviewer #4: No

---

## [Editor Report · Acceptance letter]

PONE-D-24-55413R2

PLOS One

Dear Dr. Inizan,

I'm pleased to inform you that your manuscript has been deemed suitable for publication in PLOS One. Congratulations! Your manuscript is now being handed over to our production team.

Kind regards,

on behalf of

Dr. Julian Ruiz-Saenz

Academic Editor

PLOS One